# BILINEAR MLPS ENABLE WEIGHT-BASED MECHANISTIC INTERPRETABILITY

**Michael T. Pearce**[*]
Independent
pearcemt@
alumni.stanford.edu

**Thomas Dooms**[*]
University of Antwerp
thomas.dooms@
uantwerpen.be

**Alice Rigg**
Independent
rigg.alice0@
gmail.com

**Jose Oramas**
University of Antwerp, sqIRL/IDLab
jose.oramas@uantwerpen.be

**Lee Sharkey**
Apollo Research
lee@apolloresearch.ai

## ABSTRACT

A mechanistic understanding of how MLPs do computation in deep neural networks remains elusive. Current interpretability work can extract features from hidden activations over an input dataset but generally cannot explain how MLP weights construct features. One challenge is that element-wise nonlinearities introduce higher-order interactions and make it difficult to trace computations through the MLP layer. In this paper, we analyze bilinear MLPs, a type of Gated Linear Unit (GLU) without any element-wise nonlinearity that nevertheless achieves competitive performance. Bilinear MLPs can be fully expressed in terms of linear operations using a third-order tensor, allowing flexible analysis of the weights. Analyzing the spectra of bilinear MLP weights using eigendecomposition reveals interpretable low-rank structure across toy tasks, image classification, and language modeling. We use this understanding to craft adversarial examples, uncover overfitting, and identify small language model circuits directly from the weights alone. Our results demonstrate that bilinear layers serve as an interpretable drop-in replacement for current activation functions and that weight-based interpretability is viable for understanding deep-learning models.

## 1 INTRODUCTION

Multi-layer perceptrons (MLPs) are an important component of many deep learning models, including transformers (Vaswani et al., 2017). Unfortunately, element-wise nonlinearities obscure the relationship between weights, inputs, and outputs, making it difficult to trace a neural network's decision-making process. Consequently, MLPs have previously been treated as undecomposable components in interpretability research (Elhage et al., 2021).

While early mechanistic interpretability literature explored neural network weights (Olah et al., 2017; 2020; Voss et al., 2021; Elhage et al., 2021), activation-based approaches dominate contemporary research (Petsiuk et al., 2018; Ribeiro et al., 2016; Simonyan et al., 2014; Montavon et al., 2018). In particular, most recent studies on transformers use sparse dictionary learning (SDL) to decompose latent representations into an overcomplete basis of seemingly interpretable atoms (Cunningham et al., 2024; Bricken et al., 2023b; Marks et al., 2024; Dunefsky et al., 2024). However, SDL-based approaches only describe which features are present, not how they are formed or what their downstream effect is. Previous work has approximated interactions between latent dictionaries using linear and gradient-based attribution (Marks et al., 2024; Ge et al., 2024), but these approaches offer weak guarantees of generalization. To ensure faithfulness, ideally, we would be able to capture nonlinear feature interactions in circuits that are grounded in the model weights.

---

[*]Equal contribution
Code at: https://github.com/tdooms/bilinear-decomposition

One path to better circuit discovery is to use more inherently interpretable architectures. Previous work constrains the model into using human-understandable components or concepts (Koh et al., 2020; Chen et al., 2019), but this typically requires labeled training data for the predefined concepts and involves a trade-off in accuracy compared to learning the best concepts for performance (Henighan et al., 2023). Ideally, we could more easily extract and interpret the concepts that models naturally learn rather than force the model to use particular concepts. To this end, Sharkey (2023) suggested that bilinear layers Lin et al. (2015); Li et al. (2017); Chrysos et al. (2021) of the form $g(\boldsymbol{x}) = (\boldsymbol{W}\boldsymbol{x}) \odot (\boldsymbol{V}\boldsymbol{x})$ are intrinsically interpretable because their computations can be expressed in terms of linear operations with a third order tensor. This enables the use of tensor or matrix decompositions to directly understand the weights. Moreover, bilinear layers outperform ReLU-based transformers in language modeling (Shazeer, 2020) and have performance only slightly below SwiGLU, which is prevalent in competitive models today (Touvron et al., 2023).

Tensor decompositions have long been studied in machine learning (Cichocki et al., 2015; Panagakis et al., 2021; Sidiropoulos et al., 2017) where most applications are based on an input dataset. Using decompositions to extract features directly from the weights of tensor-based models is less explored. Here, we show that bilinear MLPs can be decomposed into functionally relevant, interpretable components by directly decomposing the weights, without using inputs. These decompositions reveal a low-rank structure in bilinear MLPs trained across various tasks. In summary, this paper demonstrates that bilinear MLPs are an interpretable drop-in replacement for ordinary MLPs in a wide range of settings. Our contributions are as follows:

1. In section 3, we introduce several methods to analyze bilinear MLPs. One method decomposes the weights into a set of eigenvectors that explain the outputs along a given set of directions in a way that is fully equivalent to the layer's original computations.

2. In section 4, we showcase the eigenvector decomposition across multiple image classification tasks, revealing an interpretable low-rank structure. Smaller eigenvalue terms can be truncated while preserving performance. Using the eigenvectors, we see how regularization reduces signs of overfitting in the extracted features and construct adversarial examples.

3. Finally, in section 5, we analyze how bilinear MLPs compute output features from input features, both derived from sparse dictionary learning (SDL). We highlight a small circuit that flips the sentiment of the next token if the current token is a negation ("not"). We also find that many output features are well-correlated with low-rank approximations. This gives evidence that weight-based interpretability can be viable in large language models.

## 2    BACKGROUND

Throughout, we use conventional notation as in Goodfellow et al. (2016). Scalars are denoted by $s$, vectors by $\boldsymbol{v}$, matrices by $\boldsymbol{M}$, and third-order tensors by $\mathsf{T}$. The entry in row $i$ and column $j$ of a matrix $\boldsymbol{M}$ is a scalar and therefore denoted as $m_{ij}$. We denote taking row $i$ or column $j$ of a matrix by $\boldsymbol{m}_{i:}$ and $\boldsymbol{m}_{:j}$ respectively. We use $\odot$ to denote an element-wise product and $\cdot_{\text{axis}}$ to denote a product of tensors along the specified axis.

**Defining bilinear MLPs.** Modern Transformers (Touvron et al., 2023) feature Gated Linear Units (GLUs), which offer a performance gain over standard MLPs for the same number of parameters (Shazeer, 2020; Dauphin et al., 2017). GLU activations consist of the component-wise product of two linear up-projections of size $(d_{\text{hidden}}, d_{\text{input}})$, $\boldsymbol{W}$ and $\boldsymbol{V}$, one of which is passed through a nonlinear activation function $\sigma$(Equation 1). The hidden activations $g(\boldsymbol{x})$ then pass through a down-projection $\boldsymbol{P}$ of size $(d_{\text{output}}, d_{\text{hidden}})$. We omit biases for brevity.

$$g(\boldsymbol{x}) = (\boldsymbol{W}\boldsymbol{x}) \odot \sigma(\boldsymbol{V}\boldsymbol{x})$$
$$\text{GLU}(\boldsymbol{x}) = \boldsymbol{P}(g(\boldsymbol{x})) \tag{1}$$

A bilinear layer is a GLU variant that omits the nonlinear activation function $\sigma$. Bilinear layers beat ordinary ReLU MLPs and perform almost as well as SwiGLU on language modeling tasks (Shazeer, 2020). We corroborate these findings in Appendix I, and show that bilinear layers achieve equal loss when keeping training time constant and marginally worse loss when keeping data constant.

**Interaction matrices and the bilinear tensor.** A bilinear MLP parameterizes the pairwise interactions between inputs. One way to see this is by looking at how a single output $g(\mathbf{x})_a$ is computed.

$$g(\boldsymbol{x}) = (\boldsymbol{W}\boldsymbol{x}) \odot (\boldsymbol{V}\boldsymbol{x})$$
$$g(\boldsymbol{x})_a = (\boldsymbol{w}_{a:}^T \boldsymbol{x})\,(\boldsymbol{v}_{a:}^T \boldsymbol{x})$$
$$= \boldsymbol{x}^T (\boldsymbol{w}_{a:}\boldsymbol{v}_{a:}^T)\boldsymbol{x}$$

We call the $(d_{\text{input}}, d_{\text{input}})$ matrix $\boldsymbol{w}_{a:}\boldsymbol{v}_{a:}^T = \boldsymbol{B}_{a::}$ an *interaction matrix* since it defines how each pair of inputs interact for a given output dimension $a$.

The collection of interaction matrices across the output axis can be organized into the third-order *bilinear tensor*, **B**, with elements $b_{aij} = w_{ai}v_{aj}$, illustrated in Figure 1A. The bilinear tensor allows us to easily find the interaction matrix for a specific output direction $\boldsymbol{u}$ of interest by taking a product along the output axis, $\boldsymbol{u} \cdot_{\text{out}} \boldsymbol{B}$, equal to a weighted sum over the neuron-basis interaction matrices, $\sum_a u_a \boldsymbol{w}_{a:}\boldsymbol{v}_{a:}^T$. As written, **B** has size $(d_{\text{hidden}}, d_{\text{input}}, d_{\text{input}})$ but we will typically multiply the down-projection $\boldsymbol{P}$ into **B** resulting in a $(d_{\text{output}}, d_{\text{input}}, d_{\text{input}})$ size tensor.

**Simplifications due to symmetry.** Because an interaction matrix is always evaluated with two copies of the input $\boldsymbol{x}$, it contains redundant information that does not contribute to the activation. Any square matrix can be expressed uniquely as the sum of a symmetric and anti-symmetric matrix.

$$\boldsymbol{B}_{a::} = \frac{1}{2}\underbrace{(\boldsymbol{B}_{a::} + \boldsymbol{B}_{a::}^T)}_{\boldsymbol{B}_{a::}^{sym}} + \frac{1}{2}\underbrace{(\boldsymbol{B}_{a::} - \boldsymbol{B}_{a::}^T)}_{\boldsymbol{B}_{a::}^{anti}}$$

However, evaluating an anti-symmetric matrix $\boldsymbol{A}$ with identical inputs yields 0 and can be omitted:

$$\boldsymbol{x}^T \boldsymbol{A}\boldsymbol{x} = \boldsymbol{x}^T(-\boldsymbol{A}^T)\boldsymbol{x} = -(\boldsymbol{x}^T \boldsymbol{A}\boldsymbol{x})^T = 0.$$

Therefore, only the symmetric part $\boldsymbol{B}_{a::}^{sym}$ contributes. From here on, we drop the $\cdot^{sym}$ superscript and assume the symmetric form for any interaction matrix or bilinear tensor ($b_{aij} = \frac{1}{2}(w_{ai}v_{aj} + w_{aj}v_{ai})$). Symmetric matrices have simpler eigendecompositions since the eigenvalues are all real-valued, and the eigenvectors are orthogonal by the spectral theorem.

**Incorporating biases.** If a bilinear layer has biases, we can augment the weight matrices to adapt our approach. Given activations of the form $g(\boldsymbol{x}) = (\boldsymbol{W}\boldsymbol{x}+\boldsymbol{b}_1)\odot(\boldsymbol{V}\boldsymbol{x}+\boldsymbol{b}_2)$, define $\boldsymbol{W}' = [\boldsymbol{W};\boldsymbol{b}_2]$, $\boldsymbol{V}' = [\boldsymbol{V};\boldsymbol{b}_2]$, and $\boldsymbol{x}' = [\boldsymbol{x},1]$. Then, $g(\boldsymbol{x}) = (\boldsymbol{W}'\boldsymbol{x}') \odot (\boldsymbol{V}'\boldsymbol{x}')$ in a bilinear layer with biases. In subsection 4.3, we study a toy classification task using a model trained with biases, illustrating how biases can be interpreted using the same framework. For the rest of our experiments, we used models without biases for simplicity, as it did not harm performance. See Appendix L for details.

## 3 ANALYSIS METHODS

Since bilinear MLPs can be expressed in terms of a third-order tensor, **B**, they can be flexibly analyzed using techniques from linear algebra, such as decompositions and transformations. The choice of analysis approach depends on what additional information, in terms of previously obtained input or output features, is provided.

### 3.1 INPUT / OUTPUT FEATURES → DIRECT INTERACTIONS

If we have already obtained meaningful sets of features for the bilinear MLP's inputs and outputs, for example from a set of latent feature dictionaries $F^{\text{in}}$ and $F^{\text{out}}$, then we can directly study the interactions between these features and understand how the output features are constructed from input ones. We can transform the bilinear tensor into the feature basis via $\tilde{b}_{abc} = \sum_{ijk} f_{ai}^{\text{out}}\, b_{ijk}\, f_{bj}^{\text{in}}\, f_{ck}^{\text{in}}$. For a given set of sparse input and output activations, only a small subset of the interactions (with $a, b, c$ all active) will contribute, and the statistics of these active interactions can be studied.

For dictionaries obtained from sparse autoencoders (SAEs) we can instead use the output SAE's encoder directions in the transformation: $\tilde{b}_{abc} = \sum_{ijk} e_{ai}^{\text{out}}\, b_{ijk}\, f_{bj}^{\text{in}}\, f_{ck}^{\text{in}}$. Then the output activations are $z_c^{\text{out}} = \text{ReLU}(\sum_{ab} \tilde{b}_{abc} z_a^{\text{in}} z_b^{\text{in}})$ in terms of the input directions $\boldsymbol{z}^{\text{in}}$. In section 5, we use this approach to identify the top relevant interactions for features in a language model.

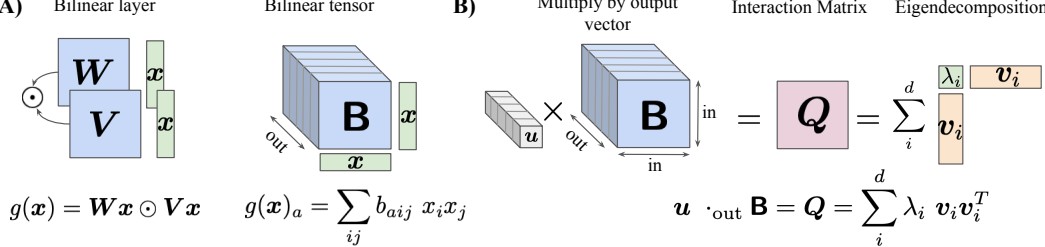

$$g(\boldsymbol{x}) = \boldsymbol{W}\boldsymbol{x} \odot \boldsymbol{V}\boldsymbol{x} \qquad g(\boldsymbol{x})_a = \sum_{ij} b_{aij}\, x_i x_j \qquad \boldsymbol{u} \cdot_{\text{out}} \mathbf{B} = \boldsymbol{Q} = \sum_i^d \lambda_i\, \boldsymbol{v}_i \boldsymbol{v}_i^T$$

Figure 1: A) Two ways to represent a bilinear layer, via an elementwise product or the bilinear tensor. B) Diagram of the eigendecomposition technique. Multiplying the bilinear tensor by a desired output direction $\boldsymbol{u}$ produces an interaction matrix $\boldsymbol{Q}$ that can be decomposed into a set of eigenvectors $\boldsymbol{v}$ and associated eigenvalues $\lambda_i$.

### 3.2 OUTPUT FEATURES → EIGENDECOMPOSITION

Given a set of meaningful features for the MLP outputs, we can identify the most important input directions that determine the output feature activations. The output features could come from a dictionary, from the unembedding (shown in section 4), or from the decompilation of later layers.

The interaction matrix, $\boldsymbol{Q} = \boldsymbol{u} \cdot_{\text{out}} \mathbf{B}$ for a given output feature $\boldsymbol{u}$ can be decomposed into a set of eigenvectors (Figure 1). Since $\boldsymbol{Q}$ can be considered symmetric without loss of generality (see section 2), the spectral theorem gives

$$\boldsymbol{Q} = \sum_i^d \lambda_i\, \boldsymbol{v}_i \boldsymbol{v}_i^T \tag{2}$$

with a set of $d$ (the rank of $\boldsymbol{W}, \boldsymbol{V}$) orthonormal eigenvectors $\boldsymbol{v}_i$ and real-valued eigenvalues $\lambda_i$. In the eigenvector basis, the output in the $\boldsymbol{u}$-direction is

$$\boldsymbol{x}^T \boldsymbol{Q} \boldsymbol{x} = \sum_i^d \underbrace{\lambda_i \left(\boldsymbol{v}_i^T \boldsymbol{x}\right)^2}_{\text{activation for } \boldsymbol{v}_i} \tag{3}$$

where each term can be considered the activation for the eigenvector $\boldsymbol{v}_i$ of size $(d_{\text{input}})$. That is, the bilinear layer's outputs are quadratic in the eigenvector basis.

The eigenvector basis makes it easy to identify any low-rank structure relevant to $\boldsymbol{u}$. The top eigenvectors by eigenvalue magnitude give the best low-rank approximation to the interaction matrix $\boldsymbol{Q}$ for a given rank. And since the eigenvectors diagonalize $\boldsymbol{Q}$, there are no cross-interactions between eigenvectors that would complicate the interpretation of their contributions to $\boldsymbol{u}$.

### 3.3 NO FEATURES → HIGHER-ORDER SVD

If we have no prior features available, it is still possible to determine the most important input and output directions of $\mathbf{B}$ through a higher-order singular value decomposition (HOSVD). The simplest approach that takes advantage of the symmetry in $\mathbf{B}$ is to reshape the tensor by flattening the two input dimensions to produce a $(d_{\text{output}}, d_{\text{input}}^2)$ shaped matrix and then do a standard singular value decomposition (SVD). Schematically, this gives

$$\boldsymbol{B}_{\text{out,in}\times\text{in}} = \sum_i \sigma_i\, \boldsymbol{u}_{\text{out}}^{(i)} \otimes \boldsymbol{q}_{\text{in}\times\text{in}}^{(i)}$$

where $\boldsymbol{q}$ can still be treated as an interaction matrix and further decomposed into eigenvectors as described above. We demonstrate this approach for an MNIST model in Appendix D.

## 4 IMAGE CLASSIFICATION: INTERPRETING VISUAL FEATURES

We consider models trained on the MNIST dataset of handwritten digits and the Fashion-MNIST dataset of clothing images. This is a semi-controlled environment that allows us to evaluate the

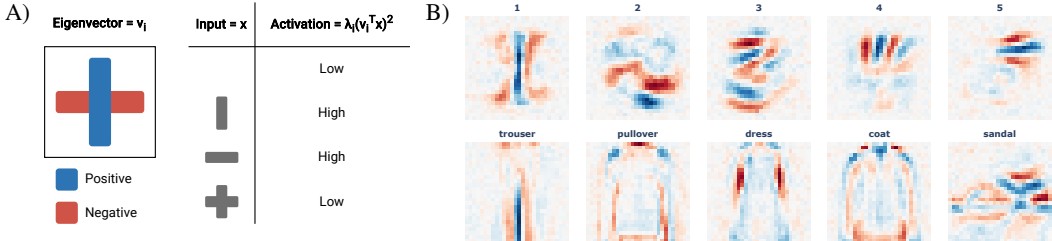

Figure 2: A) Eigenvector activations are quadratic in the input and have a large magnitude if an input aligns with the positive (blue) regions *or* the negative (red) regions, but not both. B) Top eigenvectors for single-layer MNIST and Fashion-MNIST models, revealing the most significant patterns learned for each class. In MNIST, eigenvectors represent components of the target class, while Fashion-MNIST eigenvectors function as localized edge detectors. Best viewed in color.

interpretability of eigenvectors computed using the methods in subsection 3.2. This section analyses a shallow feedforward network (FFN) consisting of an embedding projection, a bilinear layer, and a classification head; see Appendix G for details.

First, we qualitatively survey the eigenvectors and highlight the importance of regularization in feature quality. Second, we consider the consistency of eigenvectors across training runs and sizes. Third, we turn toward an algorithmic task on MNIST, where we compare the ground truth with the extracted eigenvectors. Lastly, we use these eigenvectors to construct adversarial examples, demonstrating their causal importance.

### 4.1 QUALITATIVE ASSESSMENT: TOP EIGENVECTORS APPEAR INTERPRETABLE

The eigenvectors are derived using the unembedding directions for the digits as the output directions $u$ to obtain interaction matrices $Q = u \cdot_{\text{out}} \mathbf{B}$ that are then decomposed following subsection 3.2. So each unembedding direction (digit) has a corresponding set of eigenvectors, although we may refer to the full collection as the eigenvectors of the layer or model.

We can visualize them by projecting them into the input space using the embedding weights. Because the activation of an eigenvector $v$ with eigenvalue $\lambda_i$ is quadratic in the input, $\lambda (v^T x)^2$, the sign of the eigenvector $v$ is arbitrary. The quadratic leads to XOR-like behavior where high overlap with an eigenvector's positive regions (blue) *or* the negative regions (red)—but not both—leads to large activation magnitude, while the overall sign is determined by the eigenvalue (Figure 2A).

For MNIST, the top positive eigenvector for each output class emphasizes a curve segment specific to its digit or otherwise resembles a prototypical class image (Figure 2B). Top eigenvectors for FMNIST function as localized edge detectors, focusing on important edges for each clothing article, such as the leg gap for trousers. The localized edge detection relies on the XOR-like behavior of the eigenvector's quadratic activation.

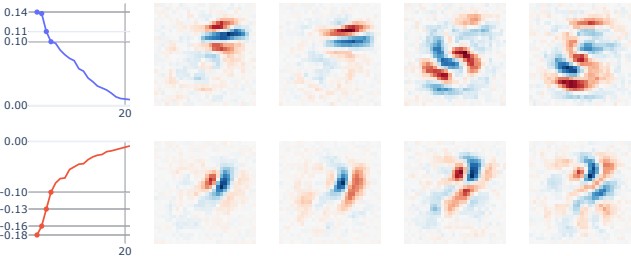

Figure 3: The top four positive (top) and negative (bottom) eigenvectors for the digit 5, ordered from left to right by importance. Their eigenvalues are highlighted on the left. Only 20 positive and 20 negative eigenvalues (out of 512) are shown on the left images. Eigenvectors tend to represent semantically and spatially coherent structures.

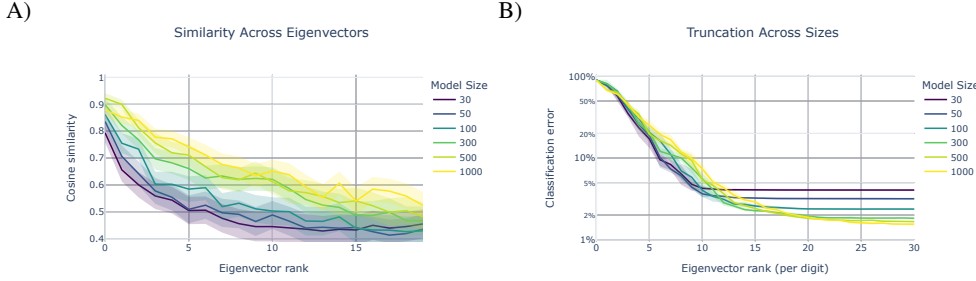

Figure 4: Top eigenvector for models trained with varying Gaussian input noise. For reference, the norm of an average digit is about 0.3; adding noise with a norm of 1 results in a heavily distorted but discernible digit. Finally, the test accuracy for each model is shown at the top.

Only a small fraction of eigenvalues have non-negligible magnitude (Figure 3). Different top eigenvectors capture semantically different aspects of the class. For example, in the spectrum for digit 5, the first two positive eigenvectors detect the 5's horizontal top stroke but at different positions, similar to Gabor filters. The next two positive eigenvectors detect the bottom segment. The negative eigenvectors are somewhat less intuitive but generally correspond to features that indicate the digit is not a five, such as an upward curve in the top right quadrant instead of a horizontal stroke. In Appendix C, we study this technique towards explaining an input prediction. Details of the training setup are outlined in Appendix G while similar plots for other digits can be found in Appendix A.

Because we can extract features directly from model weights, we can identify overfitting in image models by visualizing the top eigenvectors and searching for spatial artifacts. For instance, the eigenvectors of unregularized models focus on certain outlying pixels (Figure 4). We found adding dense Gaussian noise to the inputs (Bricken et al., 2023a) to be an effective model regularizer, producing bilinear layers with more intuitively interpretable features. Increasing the scale of the added noise results produces more digit-like eigenvectors and results in a lower-rank eigenvalue spectrum (Appendix E). These results indicate that our technique can qualitatively help uncover overfitting or other unwanted behavior in models. Furthermore, it can be used to evaluate the effect of certain regularizers and augmentation techniques, as explored in Appendix B.

## 4.2 QUANTITATIVE ASSESSMENT: EIGENVECTORS LEARN CONSISTENT PATTERNS

One important question in machine learning is whether models learn the same structure across training runs (Li et al., 2016) and across model sizes (Frankle & Carbin, 2019). In this section, we study both and find that eigenvectors are similar across runs and behave similarly across model sizes. Furthermore, we characterize the impact of eigenvector truncation on classification accuracy.

Both the ordering and contents of top eigenvectors are very consistent across runs. The cosine similarities of the top eigenvector are between 0.8 and 0.9 depending on size (Figure 5). Generally,

Figure 5: A) The similarity between ordered eigenvectors of the same model size averaged over all digits. This shows that equally sized models learn similar features. B) Resulting accuracy after only retaining the $n$ most important eigenvalues (per digit). Both plots are averaged over 5 runs with the 90% confidence interval shown.

increasing model sizes results in more similar top eigenvectors. Further, truncating all but the top few eigenvectors across model sizes yields very similar classification accuracy. This implies that, beyond being consistently similar, these eigenvectors have a comparable impact on classification. In Appendix F, we further study the similarity of eigenvectors between sizes and show that retaining only a handful of eigenvectors results in minimal accuracy drops (0.01%).

### 4.3 COMPARING WITH GROUND TRUTH: EIGENVECTORS FIND COMPUTATION

To perform a ground-truth assessment of eigenvectors, we consider a task from a mechanistic interpretability challenge, where the goal was to determine the labeling function (training objective) from a model Casper (2023). Specifically, the challenge required reverse-engineering a binary image classifier trained on MNIST, where the label is based on the similarity to a specific target image. The model predicted 'True' if the input has high cosine similarity to this target or high similarity to the complement (one minus the grayscale) of that target and 'False' otherwise. This target is chosen as an instance of a '1'.

Previous work (Stefan Heimersheim, 2023) reverse-engineered this through a combination of methods, all requiring careful consideration and consisting of non-trivial insights. Furthermore, the methods required knowledge of the original dataset and a hint of what to look for. While our method does not work on the original architecture, we show that we do not require such knowledge and can extract the original algorithm from the weights alone.

We perform our decomposition on the output difference (True − False) since this is the only meaningful direction before the softmax. This consistently reveals one high positive eigenvalue; the rest are (close to) zero (Figure 6). The most positive eigenvector is sufficient for completing the task; it computes the exact similarity we want. If the input is close to the target, the blue region will match; if it is close to the complement, the red will match; if both are active simultaneously, they will somewhat cancel out. The remaining two eigenvectors are separated as they seem to overfit the data slightly; the negative eigenvector seems to penalize diagonal structures.

Contrary to other models, this task greatly benefited from including biases. This arises from the fact that the model must not only compute similarity but also make its binary decision based on a learned threshold. If no bias is provided, the model attempts to find quadratic invariances in the data, which don't generalize well, especially given the important but sensitive role of this threshold in classification. Here, the bias (shown in the bottom corner of Figure 6) represents a negative contribution. The role of biases in bilinear layers is further discussed in Appendix L.

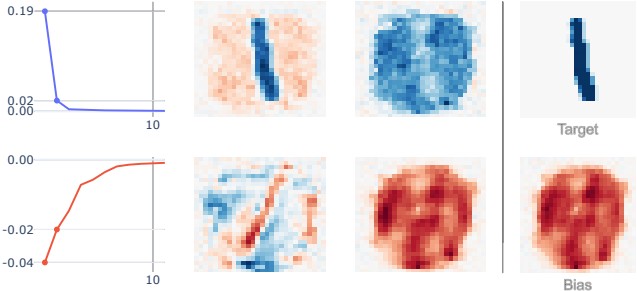

Figure 6: Eigenvalues and eigenvectors of a model trained to classify based on similarity to a target. The most important eigenvector (top-left) is a generalizing solution; the other features sharpen the decision boundary based on the training dataset. The latter features disappear with increased regularization. On the right, the target digit is shown along with the learned bias from the model.

### 4.4 ADVERSARIAL MASKS: GENERAL ATTACKS FROM WEIGHTS

To demonstrate the utility of weight-based decomposition, we construct adversarial masks for the MNIST model without training or any forward passes. These masks are added to the input, leading to misclassification as the adversarial digit. The effect is similar to steering, but the intervention is at the input instead of the model internals.

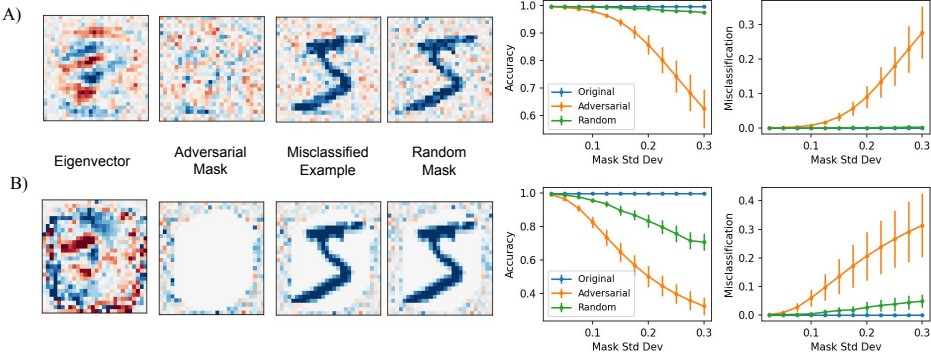

Figure 7: Examples of an adversarial mask constructed from the given eigenvector along for models trained A) with Gaussian noise regularization (std 0.15) and B) without regularization. The average accuracy and the rate of misclassification as the adversarial digit show stronger effects for adversarial masks than random baselines. In B), the mask is only applied to the outer edge of pixels that are active on less than 1% of samples.

We construct the adversarial masks from the eigenvectors for specific digits. One complication is that the eigenvectors can have nontrivial cosine similarity with each other, so an input along a single eigenvector direction could potentially activate multiple eigenvectors across different digits. To help avoid this, we construct the mask $m_i$ for a given eigenvector $v_{i:}$ as the corresponding row of the pseudoinverse $(V^+)_{i:}$ for a set of eigenvectors $V$ (specifically the top 10 positive). In an analogy to key-value pairs, the pseudoinverses effectively act like keys that activate with more specificity than the eigenvectors themselves, since $v_{j:} \cdot (V^+)_{i:} = \delta_{ij}$.

We construct an adversarial mask from an eigenvector for the digit 3 (Figure 7A). Even though the original eigenvector resembles the digit, the pseudoinverse-based mask does not (see Appendix M for more examples). The accuracy, averaged over masks from the top three eigenvectors, drops significantly more than the baseline of randomly permuting the mask despite regularizing the model during training using dense Gaussian noise with a standard deviation of 0.15. The corresponding rise in misclassification indicates effective steering towards the adversarial digit.

Ilyas et al. (2019) observe that adversarial examples can arise from predictive but non-robust features of the data, perhaps explaining why they often transfer to other models. Our construction can be seen as a toy realization of this phenomenon because the masks correspond to directions that are predictive of robust features but are not robust. We construct masks that only exploit the patterns of over-fitting found on the outer edge of the image for a model trained without regularization (Figure 7B). Since we can find this over-fitting pattern from the eigenvectors, in a general way, we can construct the mask by hand instead of optimizing it.

# 5 LANGUAGE: FINDING INTERACTIONS BETWEEN SAE FEATURES

Each output of a bilinear layer is described by weighted pairwise interactions between their input features. Previous sections show that this can be successfully leveraged to trace between a bilinear layer's inputs and outputs. Here, we turn towards tracing between latent feature dictionaries obtained by training sparse autoencoders (SAEs) on the MLP inputs or outputs for a 6-layer bilinear transformer trained on TinyStories (Eldan & Li, 2023) (see training details in Appendix G).

## 5.1 SENTIMENT NEGATION CIRCUIT

We focus on using the eigendecomposition to identify low-rank, single-layer circuits in a bilinear transformer. We cherry-pick and discuss one such circuit that takes input sentiment features and semantically negates them. Unlike previous work on sparse feature circuits (Marks et al., 2024) that relies on gradient-based linear approximations, we identify nonlinear interactions grounded in the layer's weights that contribute to the circuit's computation.

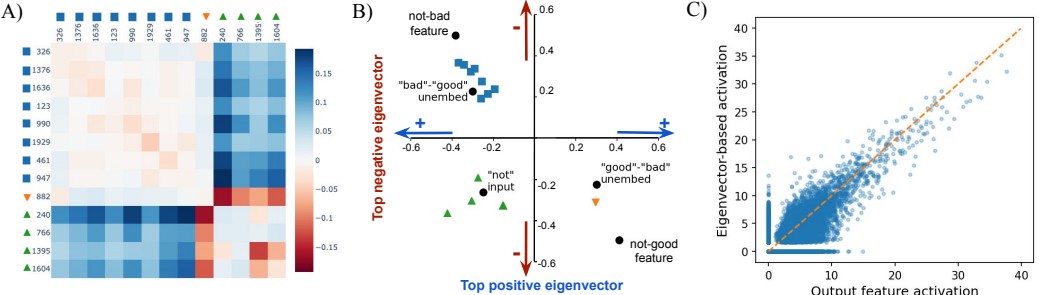

Figure 8: The sentiment negation circuit that computes the not-good and not-bad output features. A) The interaction submatrix containing the top 15 interactions. B) The projection of top interacting features onto the top eigenvectors using cosine similarity. The symbols for different clusters match the labels in A. Clusters coincide with the projection of meaningful directions such as the difference in "bad" vs "good" token unembeddings and the MLP's input activations for the input "[BOS] not". C) The not-good feature activation compared to its approximation by the top two eigenvectors.

The sentiment negation circuit computes the activation of two opposing output features in layer 4 (index 1882 and 1179) that form a fully linear subspace. The cosine similarity of their decoder vectors is -0.975. Based on their top activations, the output features activate on negation tokens ("not", "never", "wasn't") and boosts either positive sentiment tokens ("good", "safe", "nice") or negative sentiment tokens ("bad", "hurt", "sad"), so we denote the two features as the *not-good* and the *not-bad* features respectively. See Appendix O for the top activations of all features mentioned.

Focusing on the not-good feature, the top interactions for computing its activations resemble an AND-gate (Figure 8A). Input features that boost negative sentiment tokens (blue squares) have strong positive interactions with negation token features (green triangles), but both have negligible self-interactions. So, both types of input features are needed to activate the not-good feature and flip the boost from negative to positive sentiment. The one positive sentiment feature (orange downward triangle) interacts with the opposite sign. The interactions shown are significantly larger than the typical cross-interactions with a standard deviation of 0.004 (Figure 27.

The eigenvalue spectrum has one large positive (0.62) and one large negative value (-0.66) as outliers (Figure 27). We can see the underlying geometry of the circuit computation by projecting the input features onto these eigenvectors (Figure 8). By itself, a positive sentiment feature (blue squares) would equally activate both eigenvectors and cancel out, but if a negation feature is also present, the positive eigenvector is strongly activated. The activation based on only these two eigenvectors, following Equation 3, has a good correlation (0.66) with the activation of the not-good feature, particularly at large activation values (0.76), conditioned on the not-good feature being active.

## 5.2 LOW-RANK APPROXIMATIONS OF OUTPUT FEATURE ACTIVATIONS

The top eigenvectors can be used to approximate the activations of the SAE output features using a truncated form of Equation 3. To focus on the more meaningful tail of large activations, we compute the approximation's correlation conditioned on the output feature being active. The correlations of inactive features are generally lower because they are dominated by 'noise'. We evaluate this on three bilinear transformers at approximately 2/3 depth: a 6-layer TinyStories ('ts-tiny') and two FineWeb models with 12 and 16 layers ('fw-small' and 'fw-medium').

We find that features are surprisingly low-rank, with the average correlation starting around 0.65 for approximations by a single eigenvector and rising steadily with additional eigenvectors (Figure 9A). Most features have a high correlation ($> 0.75$) even when approximated by just two eigenvectors (Figure 9B). Scatter plots for a random sample of features show that the low-rank approximation often captures the tail dependence well (Figure 9C). Interestingly, we find the approximation to drastically improve with longer SAE training times while other metrics change only slightly. This indicates a 'hidden' transition near convergence and is further discussed in Appendix H. Overall, these results suggest that the interactions that produce large output activations are low-rank, making their interpretability potentially easier.

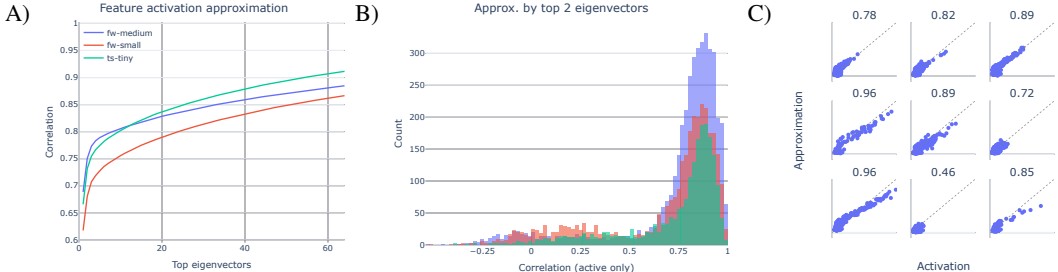

Figure 9: Activation correlations with low-rank approximations for differently-sized transformers. A) Average correlation over output features computed over every input where the feature is active. B) The distribution of active-only correlations for approximations using the top two eigenvectors. C) Scatter plots for a random set of nine output features on 'fw-medium'. Approximations use the top two eigenvectors. Low correlation scores generally only occur on low-activation features.

## 6 DISCUSSION

**Summary.** This paper introduces a novel approach to weight-based interpretability that leverages the close-to-linear structure of bilinear layers. A key result is that we can identify the most important input directions that explain the layer's output along a given direction using an eigenvector decomposition. The top eigenvectors are often interpretable, for example for MNIST they function as edge-detectors for strokes specific to each digit. The lack of element-wise nonlinearity in bilinear MLPs allows us to transform their weights into interaction matrices that connect input to output features and then extract the low-rank structure. In language models, we find that many SAE output features are well-approximated by low-rank interaction matrices, particularly at large activations. We highlighted one example of an extracted low-rank circuit that flips the sentiment of the next token if the current token is a negation ("not"). The behavior of this circuit can be easily understood in terms of the top eigenvectors, whereas finding a similar circuit in conventional MLPs would be more difficult. Overall, our results demonstrate that bilinear MLPs offer intrinsic interpretability that can aid in feature and circuit extraction.

**Implications.** The main implication of our work is that weight-based interpretability is viable, even for large language models. Bilinear MLPs can replace conventional MLPs in transformers with minimal cost while offering intrinsic interpretability due to their lack of element-wise nonlinearities and close-to-linear structure. Current circuit analysis techniques rely on gradient-based approximations (Syed et al., 2023; Marks et al., 2024) or use transcoders (Dunefsky et al., 2024) to approximate MLPs. Both approaches depend on an input dataset, potentially leading to poor performance out-of-distribution, and they may not fully capture the nonlinear computations in MLPs. In contrast, bilinear MLPs can be transformed into explicit feature interaction matrices and decomposed in a way fully equivalent to the original computations. Extracting interactions more directly from the weights should lead to better, more robust circuits. Weight-based interpretability may also offer better safety guarantees since we could plausibly prove bounds on a layer's outputs by quantifying the residual weights not captured in a circuit's interactions.

**Limitations.** Application of our methods typically relies on having a set of meaningful output directions available. In shallow models, the unembedding directions can be used, but in deeper models, we rely on features derived from sparse autoencoders that are dependent on an input dataset. Another limitation is that, although the eigenvalue spectra are often low-rank and the top eigenvectors appear interpretable, there are no guarantees the eigenvectors will be monosemantic. We expect that for high-rank spectra, the orthogonality between eigenvectors may limit their interpretability. Applying sparse dictionary learning approaches to decompose the bilinear tensor may be a promising way to relax the orthogonality constraint and find interpretable features from model weights.

## ACKNOWLEDGEMENTS

We are grateful to Narmeen Oozeer, Nora Belrose, Philippe Chlenski, and Kola Ayonrinde for helpful feedback on the draft. We are grateful to the AI Safety Camp program where this work first

started and to the ML Alignment & Theory Scholars (MATS) program that supported Michael and Alice while working on this project. We thank CoreWeave for providing compute for the finetuning experiments. This research received funding from the Flemish Government under the "Onderzoeksprogramma Artificiële Intelligentie (AI) Vlaanderen" programme.

## CONTRIBUTIONS

Michael performed the bulk of the work on the MNIST analysis and provided valuable insights across all presented topics. Thomas worked on the Language Models section and was responsible for code infrastructure. The paper was written in tandem, each focusing on their respective section.

## ETHICS STATEMENT

This paper proposes no advancements to the state-of-the-art in model capabilities. Rather, it provides new methods to analyze the internals of models to increase our understanding. The only misuse the authors envision is using this technique to leak details about the dataset that the model has learned more efficiently. However, this can be avoided by using this technique during safety evaluation.

## REPRODUCIBILITY STATEMENT

We aspired to make this work as reproducible as possible. First, Appendix G (among others) aims to provide detailed and sufficient descriptions to independently recreate our training setups. Second, our code (currently public but not referenced for anonymity) contains separate files that can be used to generate the figures in this paper independently. We used seeds across training runs so that recreated figures would be equivalent. Third, all models that are compute-intensive to train, such as the SAEs and the LMs, will be shared publicly. Lastly, we will publish an interactive demo, which will allow independent analysis of the figures in Appendix A, Appendix B, and Appendix O in a way this document cannot.

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

## A    EIGENSPECTRA: SHOWING EIGENVECTORS ACROSS DIGITS

The following are plots showing multiple positive and negative eigenvectors for certain digits. Positive features either tend to look for specific patterns in the target class (first eigenvector of 2, matching the bottom part) or tend to match an archetypal pattern (second eigenvector of 6, matching the whole digit). Negative eigenvectors tend to look for a specific part that would change the class of the digit. For instance, if the pattern highlighted by the first negative eigenvector of 4 were on, it would most likely be a 9.

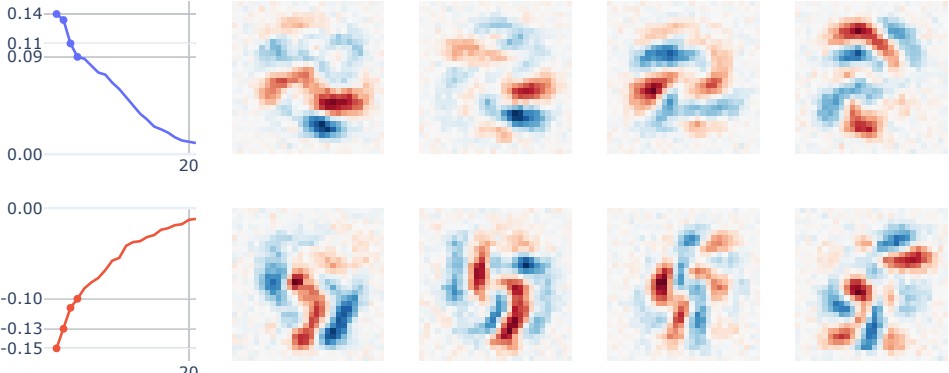

Figure 10: eigenvectors for digit 2.

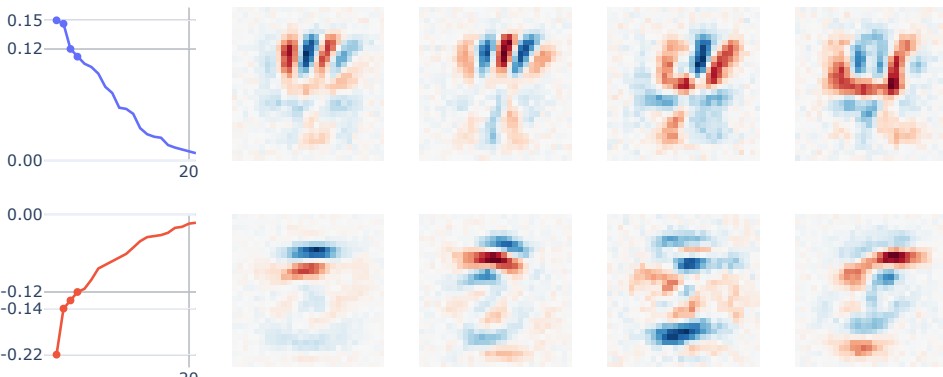

Figure 11: eigenvectors for digit 4.

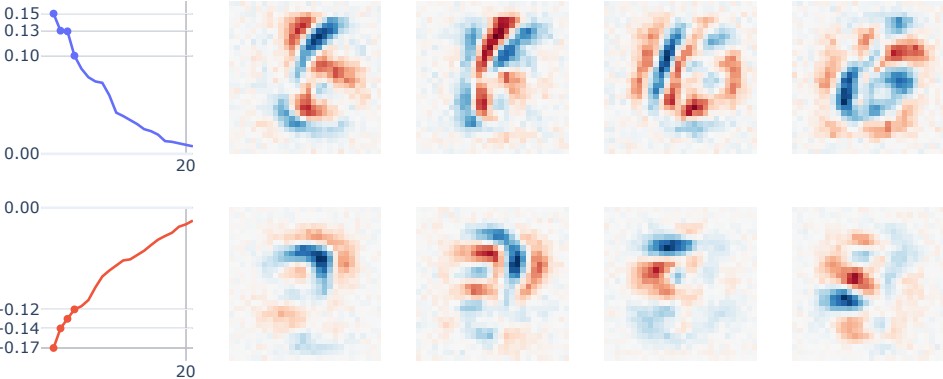

Figure 12: eigenvectors for digit 6.

## B    REGULARIZATION & AUGMENTATION: ABLATIONS & OBSERVATIONS

Following the observation that regularization improves feature interpretability for image classifiers, we study several popular regularization and augmentation techniques. In summary, input noise sparsifies the features, while geometric transformations blur the features. Some popular techniques, such as dropout, have little impact on features.

### B.1    REGULARIZATION

**Input noise** has the largest impact on features from any of the explored techniques. Specifically, we found dense Gaussian noise (already depicted in Figure 4) to provide the best trade-off between feature interpretability and accuracy. We also considered sparse salt-and-pepper noise (blacking or whiting out pixels), which resulted in both lower accuracy and interpretability. Lastly, we explored Perlin noise, which is spatially correlated and produces smooth patches of perturbation. However, this performed worst of all, not fixing the overfitting.

**Model noise** adds random Gaussian noise to the activations. This had no measurable impact on any of our experiments. However, this may simply be because our models are quite shallow.

**Weight decay** generally acts as a sparsifier for eigenvalues but does not significantly impact the eigenvectors. This is extremely useful as it can zero out the long tail of unimportant eigenvalues, strongly reducing the labor required to analyze a model fully (more details in Appendix E).

**Dropout** did not seem to impact our models. Overfitting was still an issue, even for very high values ($> 0.5$). We suspect this may change in larger or capacity-constrained models.

### B.2    AUGMENTATION

**Translation** stretches features in all directions, making them smoother. The maximal shift (right) is about 7 pixels in each direction, which is generally the maximal amount without losing important information. Interestingly, translation does not avoid overfitting but rather results in smoother overfitting patches. High translation results in split features, detecting the same pattern in different locations (Figure 13). This also results in a higher rank.

**Rotation** affects the features in the expected manner. Since rotating the digit zero does not significantly impact features, we consider the digit 5, which has a mix of rotation invariance and variance. Again, it does not stop the model from learning overfitting patches near the edges without noise. These features become broader with increased rotation.

**Blur** does not significantly affect features beyond making them somewhat smoother. Again, it still overfits certain edge pixels in a blurred manner without noise.

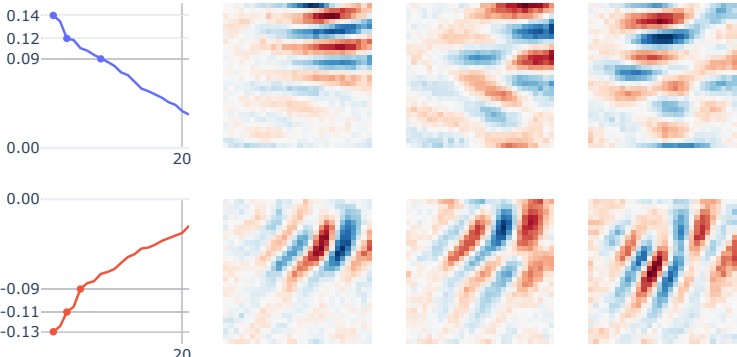

Figure 13: Important eigenvectors for a model trained with high translation regularization (7 pixels on either side). Similar patterns manifest as multiple eigenvectors at different locations.

All these augmentations are shown separately in Figure 14. Combining augmentations has the expected effect. For instance, blurring and rotation augmentation yield smooth and curvy features.

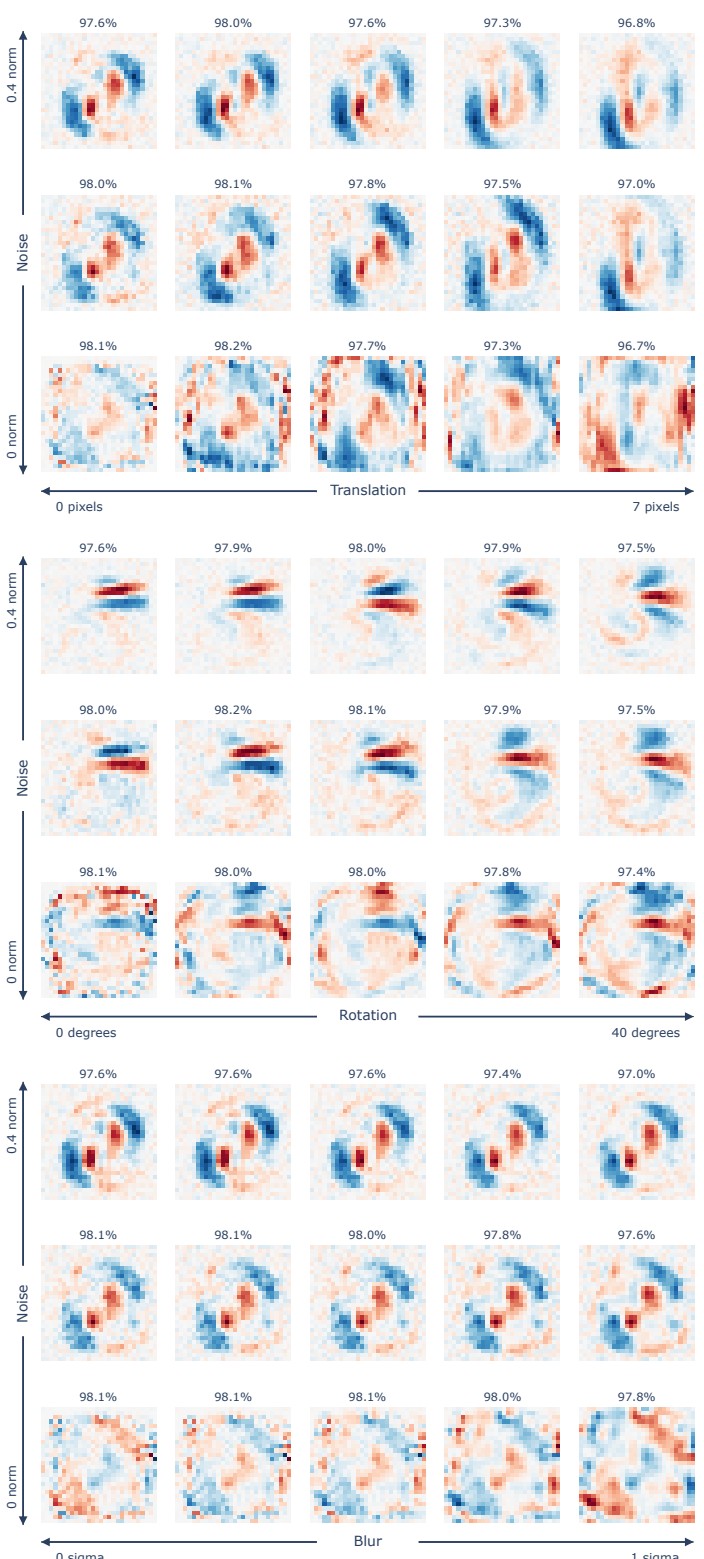

Figure 14: Important eigenvectors for models with different hyperparameters.

## C    EXPLAINABILITY: A SMALL CASE STUDY WITH EIGENVECTORS

While this paper focuses on bilinear layers for interpretability, the proposed techniques can also be used for post-hoc explainability to understand what has gone wrong. Our explanations are generally not as human-friendly as other methods but are fully grounded in the model's weights. This section explores explaining two test-set examples, one correctly classified and one incorrectly.

The figures are divided into three parts. The left line plots indicate the sorted eigenvector activation strengths for the digits with the highest logits. The middle parts visualize the top positive and negative eigenvectors for each digit. The right displays the input under study and the related logits.

The first example, a somewhat badly drawn five, results in about equal positive activations for the output classes 5, 6, and 8 (which all somewhat match this digit). The negative eigenvectors are most important in this classification, where class 5 is by far the least suppressed. This is an interesting example of the model correctly classifying through suppression.

The second example, a seven-y looking two, is actually classified as a 7. From looking at the top eigenvectors of the digit 2 (shown in Figure 10), we see that the more horizontal top stroke and more vertical slanted stroke activates the top eigenvector for the digit 7 more strongly than the 2-eigenvectors that look for more curved and slanted strokes. The negative eigenvectors are not very important in this incorrect classification.

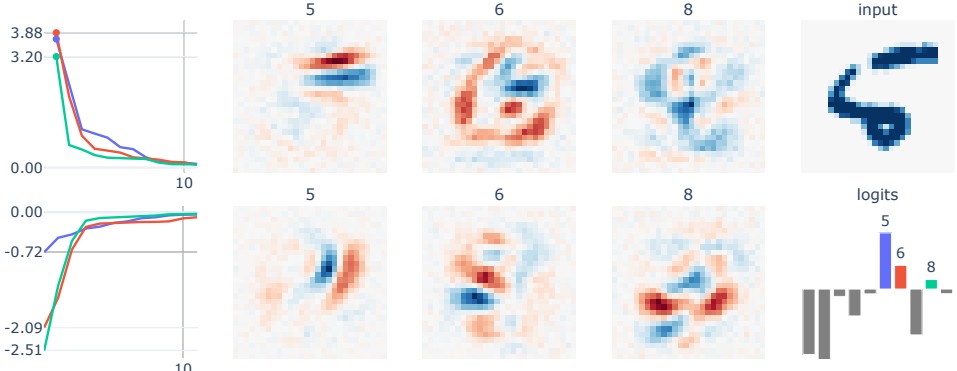

Figure 15: Study of a correctly classified 5. The output is strongly influenced by negative eigenvectors, resulting in strong suppression for the other digits.

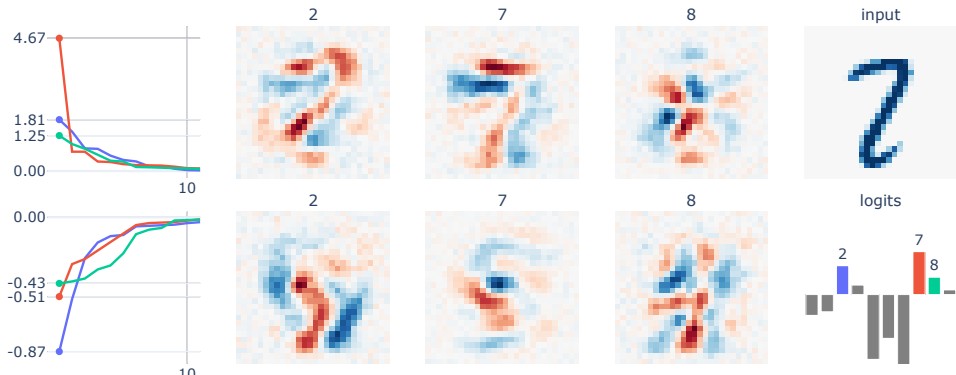

Figure 16: Study of a misclassified 2. The model mostly classifies twos based on the bottom line and top curve, which are both only partially present.

## D  HOSVD: FINDING THE MOST IMPORTANT SHARED FEATURES

In the case that no output features are available or we wish to find the dominant output directions, we can use HOSVD on the **B** tensor (described in subsection 3.3). Intuitively, this reveals the most important shared features. We demonstrate this approach on the same MNIST model used in section 4.

Instead of contributing to a single output dimension, each interaction matrix can contribute to an arbitrary direction, shown at the bottom right ("contributions"). Further, the importance of the contributions is determined by their singular value, which is shown at the top right. The remainder of the visualization shows the top eigenvalues and the corresponding eigenvectors.

The most important output direction separates digits with a prominent vertical line (1, 4, and 7) from digits with a prominent horizontal line (5 specifically). Similarly, the second most important direction splits horizontal from vertical lines but is more localized to the top half. Specifically, it splits by the orientation of the top stroke (whether it starts/ends left or right).

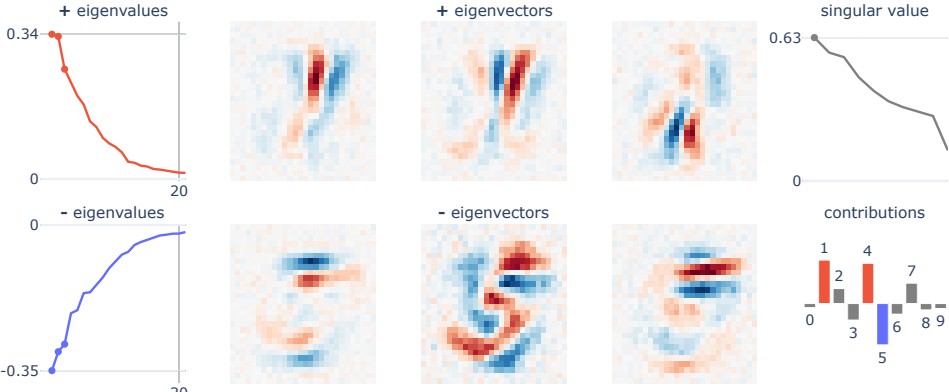

Figure 17: The most important output direction of an MNIST model roughly splits digits by its horizontality or verticality.

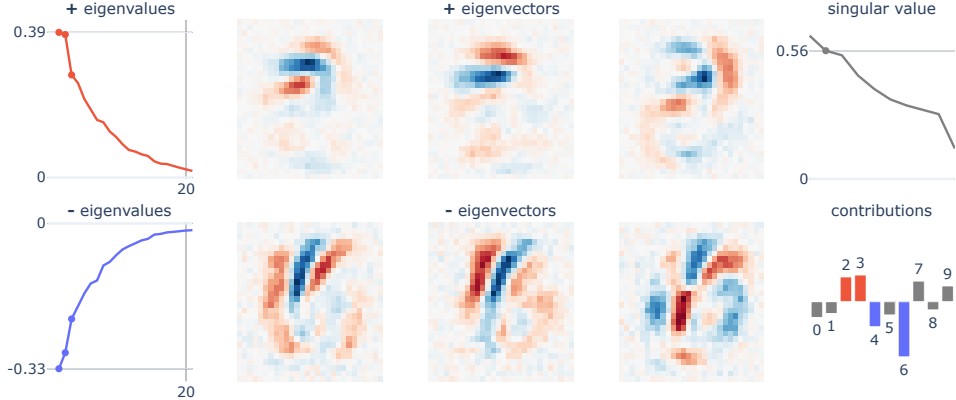

Figure 18: The second most important output direction of an MNIST model splits digits according to the orientation of its top stroke.

We observe that the output directions uncovered through HOSVD somewhat correspond to meaningful concepts, albeit sometimes dominated by a specific digit (such as 5 and 6). Less significant directions often highlight specific portions of digits that seem meaningful but are more challenging to describe. In summary, while in the case of MNIST, the results are not particularly more interpretable than decomposing according to digits, we believe this technique increases in utility (but also computational cost) as the number of output classes increases.

# E SPARSITY: WEIGHT DECAY VERSUS INPUT NOISE

Throughout, we make the claims that input noise helps create cleaner eigenvectors and that weight decay results in lower rank; this appendix aims to quantify these claims. To quantify near-sparsity, we use $(L_1/L_2)^2$, which can be seen as a continuous version of the $L_0$ norm, accounting for near-zero values. We analyze both the eigenvalues, indicating the effective rank, and the top eigenvectors, indicating the effective pixel count.

As visually observed in Figure 4, this analysis (left of Figure 19) shows that input noise plays a large role in determining the eigenvector sparsity; weight decay does not. On the other hand, input noise increases the number of important eigenvectors while weight decay decreases it. Intuitively, input noise results in specialized but more eigenvectors, while weight decay lowers the rank.

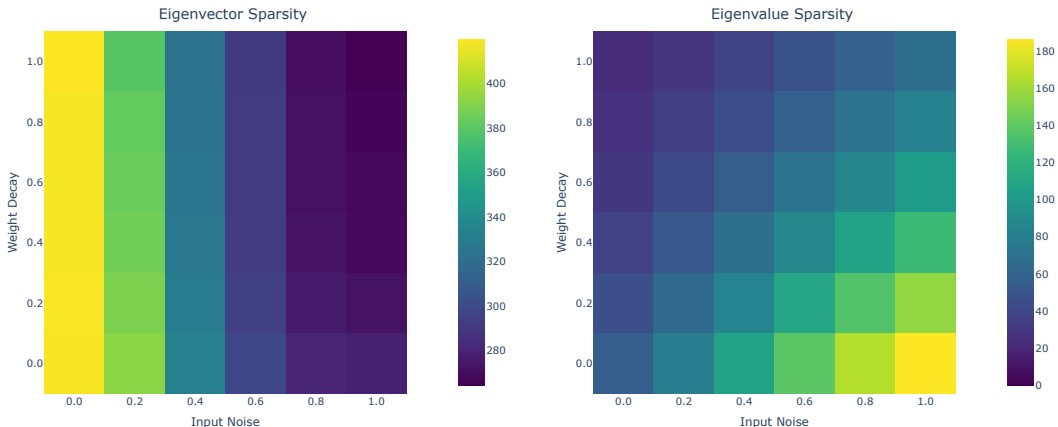

Figure 19: Measuring the approximate $L_0$ norm of eigenvalues (left) and the top 5 eigenvectors (right) with varying Gaussian input noise and weight decay.

# F TRUNCATION & SIMILARITY: A COMPARISON ACROSS SIZES

This appendix contains a brief extension of the results presented in Figure 5.

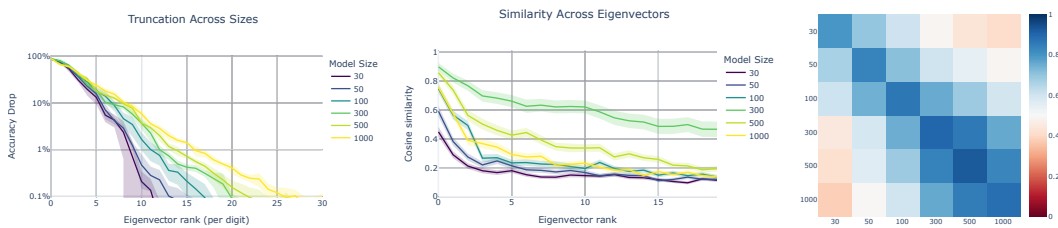

Figure 20: A) The accuracy drop when truncating the model to a limited number of eigenvectors. The first few eigenvectors result in a similar drop across sizes. Narrow models tend to naturally remain more low-rank. In general, few eigenvectors are necessary to recover the full accuracy. B) Inter-model size similarity of eigenvectors (using 300 as a comparison point). The top features for similarly-sized models are mostly similar. C) A similarity comparison between all model sizes for the top eigenvector.

## G  EXPERIMENTAL SETUPS: A DETAILED DESCRIPTION

This section contains details about our architectures used and hyperparameters to help reproduce results. More information can be found in our code [currently not referenced for anonymity].

### G.1  IMAGE CLASSIFICATION SETUP

The image classification models (section 4) in this paper consist of three parts: an embedding, the bilinear layer, and the head/unembedding, as shown in Figure 21. The training hyperparameters are found in Table 1. However, since the model is small, these parameters (except input noise; Appendix B) do not affect results much.

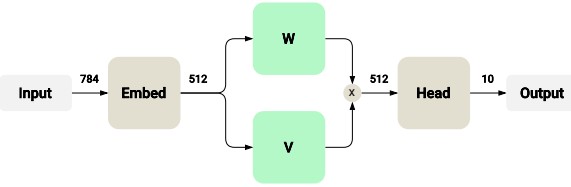

Figure 21: The architecture of the MNIST model.

| MNIST Training Parameters | |
|---|---|
| input noise norm | 0.5 |
| weight decay | 1.0 |
| learning rate | 0.001 |
| batch size | 2048 |
| optimizer | AdamW |
| schedule | cosine annealing |
| epochs | 20-100 |

Table 1: Training setup for the MNIST models, unless otherwise stated in the text.

### G.2  LANGUAGE MODEL SETUP

The language model used in section 5 is a 6-layer modern transformer model (Touvron et al., 2023) where the SwiGLU is replaced with a bilinear MLP (Figure 22). The model has about 33 million parameters. The training setup is detailed in Table 3. As the training dataset, we use a simplified and cleaned version of TinyStories (Eldan & Li, 2023) that remedies the following issues.

> **Contamination:** About 20% of stories are exact duplicates (in both train and test).

> **Corruption:** Some stories contain strange symbol sequences akin to data corruption.

Furthermore, we use a custom interpretability-first BPE tokenizer. The tokenizer is lower-case only, splits on whitespace, and has a vocabulary of only 4096 tokens.

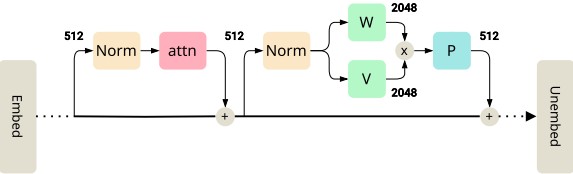

Figure 22: Simplified depiction of a bilinear transformer model. A model dimension of 512 is used, and an expansion factor of 4 (resulting in 2048 hidden dimensions in the MLP).

**TinyStories Training Parameters**

| | |
|---|---|
| **weight decay** | 0.1 |
| **batch size** | 512 |
| **context length** | 256 |
| **learning rate** | 0.001 |
| **optimizer** | AdamW |
| **schedule** | linear decay |
| **epochs** | 5 |
| **tokens** | $\pm$ 2B |
| **initialisation** | gpt2 |

Table 2: Tinystories training setup. Omitted parameters are the HuggingFace defaults.

The models used in the experiments shown in Figure 9 are trained of the FineWeb dataset (Penedo et al., 2024). These follow the architecture of GPT2-small (12 layers) and GPT2-medium (16 layers) but have bilinear MLPs. Their parameter count is 162M and 335M, respectively. Both use the Mixtral tokenizer.

**FineWeb Training Parameters**

| | |
|---|---|
| **weight decay** | 0.1 |
| **batch size** | 512 |
| **context length** | 512 |
| **learning rate** | 6e-4 |
| **optimizer** | AdamW |
| **schedule** | linear decay |
| **tokens** | $\pm$ 32B |
| **initialisation** | gpt2 |

Table 3: FineWeb training setup. Omitted parameters are the HuggingFace defaults.

### G.3 SPARSE AUTOENCODER SETUP

All discussed SAEs use a TopK activation function, as described in Gao et al. (2024). We found $k = 30$ to strike a good balance between sparseness and reconstruction loss. section 5 studies quite narrow dictionaries (4x expansion) for simplicity. The exact hyperparameters are shown in Table 4, and the attained loss added (Equation 4) across layers is shown in Figure 23.

$$L_{added} = \frac{L_{patch} - L_{clean}}{L_{clean}} \tag{4}$$

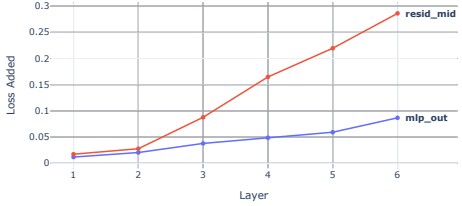

Figure 23: Loss added for the mlp_out and resid_mid SAEs across layers.

**SAE Training Parameters**

| | |
|---|---|
| **expansion** | 4x |
| **k** | 30 |
| **batch size** | 4096 |
| **learning rate** | 1e-4 |
| **optimizer** | AdamW |
| **schedule** | cosine annealing |
| **tokens** | $\pm$ 150M |
| **buffer size** | $\pm$ 2M |
| **normalize decoder** | True |
| **tied encoder init** | True |
| **encoder bias** | False |

Table 4: SAE training hyperparameters.

## H CORRELATION: ANALYZING THE IMPACT OF TRAINING TIME

Figure 9 shows how a few eigenvectors capture the essence of SAE features. This section discusses the impact of SAE quality, measured through training steps, on the resulting correlation. In short, we find features of SAEs that are trained longer are better approximated with few eigenvectors.

We train 5 SAEs with an expansion factor of 16 on the output of the MLP at layer 12 of the 'fw-medium' model. Each is trained twice as long as the last. The feature approximation correlations are computed over 100K activations; features with less than 10 activations (of which there are less than 1000) are considered dead and not shown. The reconstruction error and loss recovered between SAEs differ only by 10% while the correlation mean changes drastically (Table 5). The correlation distribution is strongly bimodal for the 'under-trained' SAEs (shown in Figure 24 with darker colors). Given more training time, this distribution shifts towards higher correlations. The activation frequencies of features are mostly uncorrelated with their approximations.

| Training steps (relative) | 1 | 2 | 4 | 8 | 16 |
|---|---|---|---|---|---|
| **Normalized MSE** | 0.17 | 0.16 | 0.16 | 0.15 | 0.15 |
| **Loss recovered** | 0.60 | 0.61 | 0.65 | 0.65 | 0.66 |
| **Correlation mean** | 0.17 | 0.28 | 0.42 | 0.52 | 0.59 |

Table 5: The SAE metrics along with the mean of the correlations shown in Figure 24. The correlation improves strongly with longer training, while other metrics only change marginally.

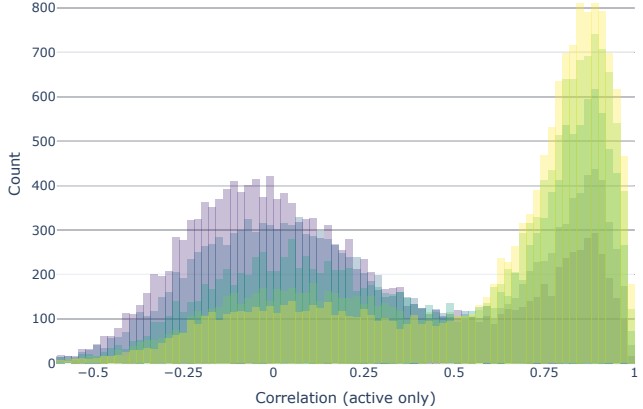

Figure 24: Feature approximation correlations using two eigenvectors across SAEs with different training times. Darker is shorter, and bright is longer. This shows a clear bimodal distribution for 'under-trained' SAEs, which vanishes upon longer training, indicating some form of convergence.

## I   BILINEAR TRANSFORMERS: A LOSS COMPARISON

While experiments on large models show bilinear layers to only marginally lag behind SwiGLU (Shazeer, 2020), this section quantifies this accuracy trade-off through compute efficiency. We performed our experiments using a 6-layer transformer model trained on TinyStories. For these experiments, we use d_model = 512 and d_hidden = 2048, resulting in roughly 30 million parameters. However, we have found these results to hold across all sizes we tried.

|                     | Bilinear | ReGLU | SwiGLU |
|---------------------|:--------:|:-----:|:------:|
| **constant epochs** | 1.337    | 1.332 | 1.321  |
| **constant time**   | 1.337    | 1.337 | 1.336  |

Table 6: The loss of language models with varying MLP activation functions. Bilinear layers are 6% less data efficient but equally compute efficient.

Considering the data efficiency (constant epochs), both SwiGLU and ReGLU marginally beat the bilinear variant. Concretely, SwiGLU attains the same final loss of the bilinear variant in 6% less epochs. On the other hand, when considering compute efficiency (constant time), we see that these differences vanish [1]. Consequently, if data is abundant, there is little disadvantage to using bilinear layers over other variants.

## J   FINETUNING: YOUR TRANSFORMER IS SECRETLY BILINEAR

Many state-of-the-art open-source models use a gated MLP called SwiGLU (Touvron et al., 2023). This uses the following activation function $\text{Swish}_\beta(x) = x \cdot \text{sigmoid}(\beta x)$. We can vary the $\beta$ parameter to represent common activation functions. If $\beta = 1$, that corresponds to SiLU activation, used by many current state-of-the-art models. $\beta = 1.7$ approximates a GELU and $\beta = 0$ is simply linear, corresponding to our setup. Consequently, we can fine-tune away the gate by interpolating $\beta$ from its original value to zero. This gradually converts an ordinary MLP into its bilinear variant.

To demonstrate how this approach performs, we fine-tuned TinyLlama-1.1B, a 1.1 billion-parameter transformer model pretrained on 3 trillion tokens of data, using a single A40 GPU. For simplicity, we trained on a slice of FineWeb data. Due to computing constraints, we only tried a single schedule that linearly interpolates towards $\beta = 0$ during the first 30% (120M tokens) and then fine-tunes for the remaining 70% (280M tokens). We compare this to a baseline that does not vary $\beta$ during fine-tuning, corresponding to continued training. We use this baseline to compensate for the difference in the pretraining distribution of TinyLlama (consisting of a mixture of RedPajama and StarCoder data). This shows that this fine-tuning process increases the loss by about (0.05) but seems to benefit from continued training (Figure 25). We plan to extend this result with a more thorough search for an improved schedule, which will probably result in a lower final loss. We also expect longer training runs to close to gap even further.

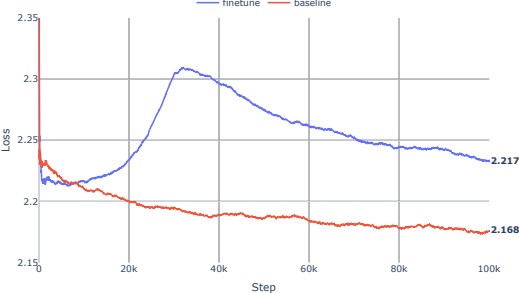

Figure 25: Comparison of fine-tuning versus a baseline over the course of 400M tokens. The loss difference is noticeable but decreases quickly with continued training.

---

[1] An improvement in implementation efficiency, such as fusing kernels, may change this fact.

## K    TENSOR DECOMPOSITIONS: EXTRACTING SHARED FEATURES

Given a complete or over-complete set of $m$ $\boldsymbol{u}$-vectors, we can re-express $\mathbf{B}$ in terms of the eigenvectors, which amounts to a change of basis. To avoid multiple contributions from similar $\boldsymbol{u}$-vectors, we have to use the pseudo-inverse, which generalizes the inverse for non-square matrices. Taking the $\boldsymbol{u}$-vectors as the columns of $\boldsymbol{U}$, the pseudo-inverse $\boldsymbol{U}^+$ satisfies $\boldsymbol{U}\boldsymbol{U}^+ = I$, as long as $\boldsymbol{U}$ is full rank (equal to $d$). Then

$$\mathbf{B} = \sum_k^m \boldsymbol{u}_{:k}^+ \otimes \boldsymbol{Q}_k \tag{5}$$

$$= \sum_k^m \sum_i^d \lambda_{\{k,i\}}\, \boldsymbol{u}_{:k}^+ \otimes \boldsymbol{v}_{\{k,i\}} \otimes \boldsymbol{v}_{\{k,i\}} \tag{6}$$

where $\boldsymbol{u}_{:k}^+$ are the rows of $U^+$ and $Q_k = \sum_i \lambda_{\{k,i\}} \boldsymbol{v}_{\{k,i\}} \otimes \boldsymbol{v}_{\{k,i\}}$ is the eigendecomposition of the interaction matrix corresponding for $\boldsymbol{u}k:$. We can then recover the interaction matrices from $\boldsymbol{Q}_k = \boldsymbol{u}_{k:}\cdot_{\text{out}} \mathbf{B}$ using the fact that $\boldsymbol{u}_{k:}\cdot\boldsymbol{u}_{:k'}^+ = \delta_{kk'}$ (Kronecker delta). Note that the eigenvectors within a single output direction $k$ are orthogonal but will overlap when comparing across different output directions.

## L    BILINEAR LAYERS: A PRACTICAL GUIDE

Bilinear layers are inherently quadratic; they can only model the importance of pairs of features, not single features. Interestingly, we haven't found this to be an issue in real-world tasks. However, linear structure is important for some toy tasks and, therefore, merits some reflection. Without modification, bilinear layers will model this linear relation as a quadratic function. To resolve this, we can add biases to the layer as follows: $\text{BL}(\boldsymbol{x}) = (\boldsymbol{W}\boldsymbol{x} + \boldsymbol{b}) \odot (\boldsymbol{V}\boldsymbol{x} + \boldsymbol{c})$. In contrast to ordinary layers, where the bias acts as a constant value, here it acts as both a linear *and* a constant value. This becomes apparent when expanded:

$$\text{BL}(\boldsymbol{x}) = \boldsymbol{W}\boldsymbol{x} \odot \boldsymbol{V}\boldsymbol{x} + (\boldsymbol{c}\boldsymbol{W}\boldsymbol{x} + \boldsymbol{b}\boldsymbol{V}\boldsymbol{x}) + \boldsymbol{c}\boldsymbol{b}$$

We disambiguate by calling the terms 'interaction', 'linear', and 'constant'. Theoretically, this is very expressive; all binary gates and most mathematical operations can be approximated quite well with it. In practice, the training process often fails to leverage this flexibility and degenerates to using quadratic invariances instead of learned constants.

## M    ADVERSARIAL MASKS: ADDITIONAL FIGURE

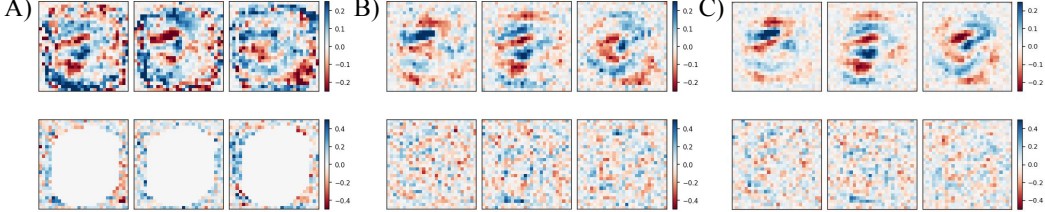

Figure 26: More examples of adversarial masks constructed from specific eigenvectors for models with A) no regularization, B) Gaussian noise regularization with std 0.15, and C) Gaussian noise regularization with std 0.3.

## N    LANGUAGE: FURTHER DETAILS FOR FEATURE CIRCUITS

A)
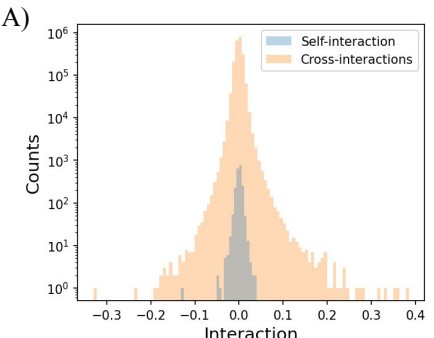
B)
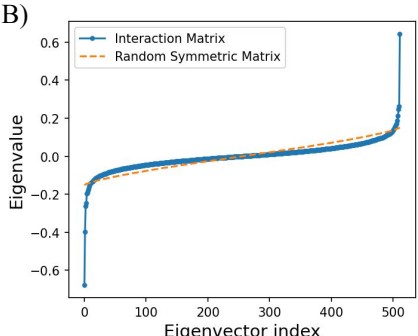

Figure 27: A) Histogram of self-interactions and cross-interactions. B) The eigenvalue spectrum for the sentiment negation feature discussed in section 5. The dashed red line gives spectrum for a random symmetric matrix with Gaussian distributed entries with the same standard deviation.

## O INPUT FEATURES OF THE NEGATION FEATURE

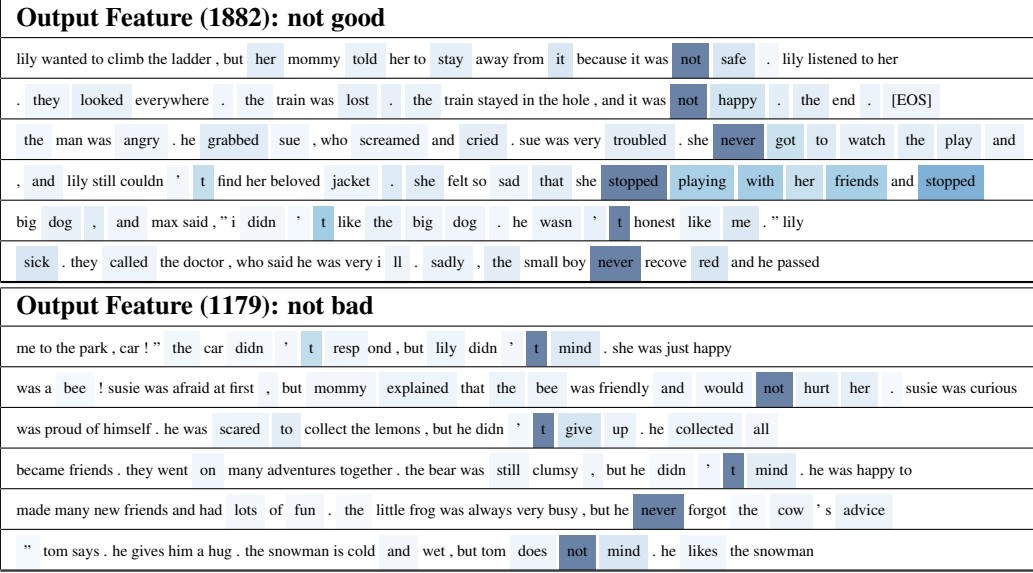

Table 7: Output SAE features in the negation feature discussed in section 5.

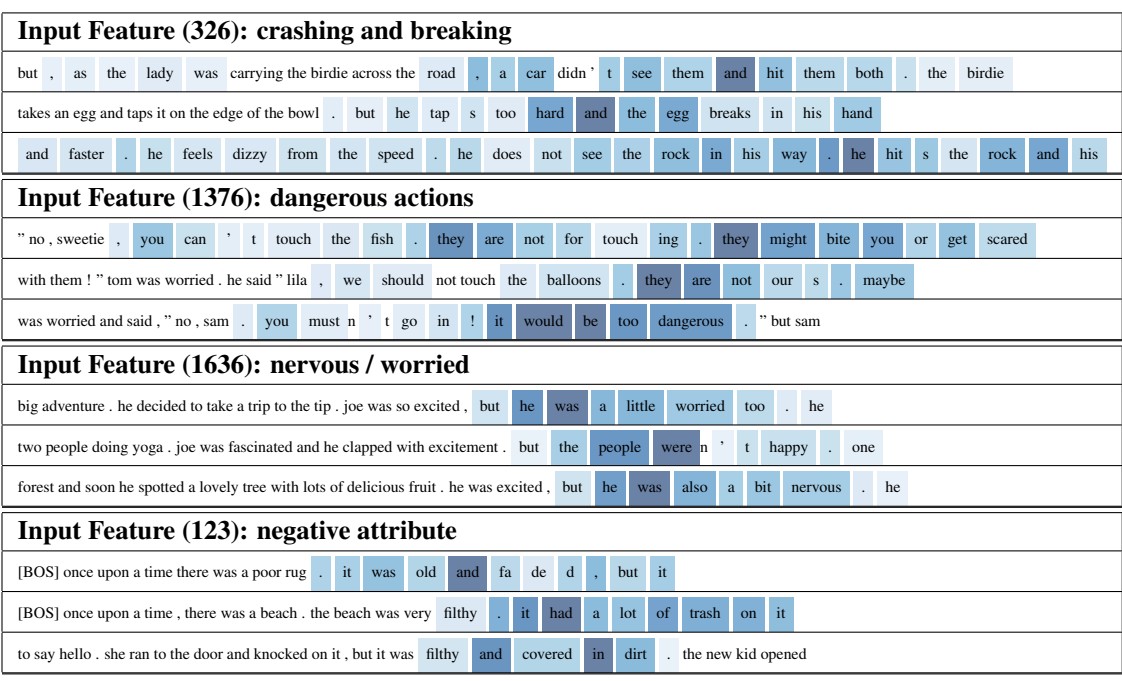

Table 8: SAE features that contribute to the negation feature discussed in section 5.

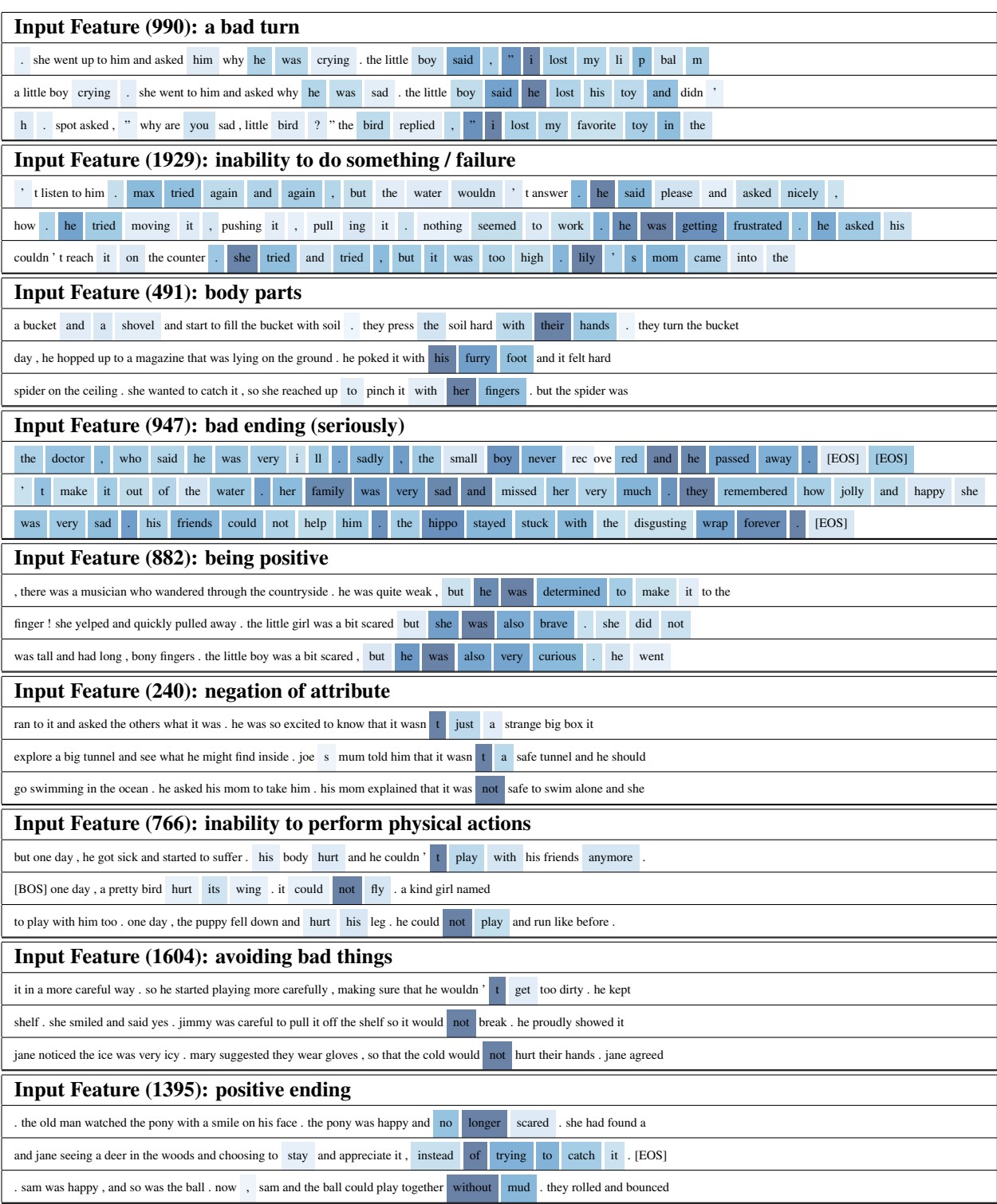

**Input Feature (990): a bad turn**

. she went up to him and asked him why he was crying . the little boy said , " i lost my li p bal m

a little boy crying . she went to him and asked why he was sad . the little boy said he lost his toy and didn '

h . spot asked , " why are you sad , little bird ? " the bird replied , " i lost my favorite toy in the

**Input Feature (1929): inability to do something / failure**

' t listen to him . max tried again and again , but the water wouldn ' t answer . he said please and asked nicely ,

how . he tried moving it , pushing it , pull ing it . nothing seemed to work . he was getting frustrated . he asked his

couldn ' t reach it on the counter . she tried and tried , but it was too high . lily ' s mom came into the

**Input Feature (491): body parts**

a bucket and a shovel and start to fill the bucket with soil . they press the soil hard with their hands . they turn the bucket

day , he hopped up to a magazine that was lying on the ground . he poked it with his furry foot and it felt hard

spider on the ceiling . she wanted to catch it , so she reached up to pinch it with her fingers . but the spider was

**Input Feature (947): bad ending (seriously)**

the doctor , who said he was very i ll . sadly , the small boy never rec ove red and he passed away . [EOS] [EOS]

' t make it out of the water . her family was very sad and missed her very much . they remembered how jolly and happy she

was very sad . his friends could not help him . the hippo stayed stuck with the disgusting wrap forever . [EOS]

**Input Feature (882): being positive**

, there was a musician who wandered through the countryside . he was quite weak , but he was determined to make it to the

finger ! she yelped and quickly pulled away . the little girl was a bit scared but she was also brave . she did not

was tall and had long , bony fingers . the little boy was a bit scared , but he was also very curious . he went

**Input Feature (240): negation of attribute**

ran to it and asked the others what it was . he was so excited to know that it wasn t just a strange big box it

explore a big tunnel and see what he might find inside . joe s mum told him that it wasn t a safe tunnel and he should

go swimming in the ocean . he asked his mom to take him . his mom explained that it was not safe to swim alone and she

**Input Feature (766): inability to perform physical actions**

but one day , he got sick and started to suffer . his body hurt and he couldn ' t play with his friends anymore .

[BOS] one day , a pretty bird hurt its wing . it could not fly . a kind girl named

to play with him too . one day , the puppy fell down and hurt his leg . he could not play and run like before .

**Input Feature (1604): avoiding bad things**

it in a more careful way . so he started playing more carefully , making sure that he wouldn ' t get too dirty . he kept

shelf . she smiled and said yes . jimmy was careful to pull it off the shelf so it would not break . he proudly showed it

jane noticed the ice was very icy . mary suggested they wear gloves , so that the cold would not hurt their hands . jane agreed

**Input Feature (1395): positive ending**

. the old man watched the pony with a smile on his face . the pony was happy and no longer scared . she had found a

and jane seeing a deer in the woods and choosing to stay and appreciate it , instead of trying to catch it . [EOS]

. sam was happy , and so was the ball . now , sam and the ball could play together without mud . they rolled and bounced

Table 9: SAE features that contribute to the negation feature discussed in section 5 (continued).

