# OpenReview forum: "Bilinear MLPs enable weight-based mechanistic interpretability"
_ICLR.cc/2025/Conference — ICLR 2025 Spotlight_

### Official Review · Reviewer_zjLd · 2024-10-20

**Soundness:** 3
**Presentation:** 2
**Contribution:** 3
**Rating:** 6
**Confidence:** 3

**Summary:**

This paper proposes to use bilinear MLP as a novel approach for weight-based interpretability. The bilinear MLP and its analysis method are introduced first. Then a handful of experiments are presented to demonstrate the usefulness of this approach, including visual feature interpretation in image classification and feature interactions in language tasks.

**Strengths:**

- This paper is carefully composed. Though there are places that can be further clarified (as mentioned below), the overall structure and logic is good.
- A variety of experiments are carried out to demonstrate the usefulness of the proposed approach. The bilinear MLP does seem to provide interesting findings and have potential in the realm of mechanistic interpretability.

**Weaknesses:**

- I couldn't understand what Figure 2 is demonstrating. Firstly, the caption refers to subfigure A, B while the figures do not have A, B labels. Secondly, do you mean that the input images with "$|$" or "$-$" patterns of exactly the same size and position will have high activation? I imagine that when flattening images to vectors, an image with many small "$-$" edges isn't well aligned with the eigenvector that is one big "$-$" in the middle of the image. If that's the case, why are the "$|$" or "$-$" eigenvectors edge detectors?
- I feel that Section 4 and 5 can be clearer if the authors always specify "eigenvectors of *what matrix*" when saying eigenvectors. Section 3 introduced three different methods. So I find the terms like "decomposition" and "eigenvectors" ambiguous, even after I tried to trace back to the concepts introduced in Section 3. The term "eigenvector of a model" is also ambiguous, since models don't really have eigenvectors.

**Questions:**

- Could the authors introduce the dimensionality of the variables when defining them? I and perhaps also other readers would find that quite useful for understanding the equations.

- In line 121, is the $W$ in $g(x)=(Wx+b_1)\odot (Vx+b_2)$ supposed to be bold?

- I'm confused by the $d_{bj}^{in}$ in line 138 and the $d_{in}$ in line 189. Are they the different variables?

- In line 252, I can see that adding dense Gaussian noise to the inputs produces bilinear layers with more intuitively interpretable features. But why is it an effective regularizer?

- Is the bilinear MLP related to the gated linear network literature [1]?

  [1] Veness et al (2021). Gated linear networks. https://arxiv.org/abs/1910.01526

---

> ### Author Response · Authors · 2024-11-16
>
> We value the reviewer's positive feedback about the technique's potential and are motivated to clarify the work as best we can. We first clarify the reviewer’s questions and then list how we included all proposed changes.
>
> > I couldn't understand what Figure 2 is demonstrating.
>
> Figure 2 provides intuition on a given eigenvector’s activation behavior. In short, the activation is defined by the squared inner product of the vector with the input. If an input matches strongly with the red (positive) or blue (negative) part of the eigenvector, the activation will be high. If it matches both, the terms for the inner product will be canceled. The activation will depend on how well the input matches each shape. If the input has a shorter stripe “-”, the activation will be lower. Next, the eigenvectors are not positionally invariant (as we don’t use CNNs), so they only match the specific indicated region. Sometimes, the model creates more spatially independent edge detectors, such as the digit 5 eigenvector, which will match with top strokes at several locations. Does the reviewer have any proposals on how to clarify the figure?
>
> > Why is input noise an effective regularizer?
>
> Training with input noise forces the model to learn robust features that can be detected despite the noise. These features tend to activate more sparsely [1] and generalize better to the test set since spurious patterns due to overfitting are less robust to noise.
>
> > Is the bilinear MLP related to the gated linear network literature?
>
> While the idea is similar, replacing non-linearities with a linear gating mechanism, the operationalization is quite different. The two main differences are that we perform weight-based interpretability by utilizing the fact that bilinear MLPs can be rewritten in terms of tensors. The halfspace gating used in GLNs is non-linear and cannot be rewritten as such. The other is that our work aims to remain as close as possible (both in spirit and accuracy) to MLPs, while their approach is quite different.
>
> To address your remaining questions/suggestions, we:
> - We have added named dimensions when introducing W, V, and B in Sec. 2
> - added A) and B) captions to Figure 2.
> - added a clarifying sentence (both in sections) about what we mean by eigenvectors (and which matrix it corresponds to).
> - changed the W on line 121 to be bold (nice catch!).
> - renamed variables on lines 138 and 189 (as they were not the same).
>
> We encourage further questions, remarks, or suggestions that would benefit this work.
>
> [1] Trenton Bricken, Rylan Schaeffer, Bruno Olshausen, and Gabriel Kreiman. Emergence of sparse representations from noise. In International Conference on Machine Learning, pp. 3148–3191. PMLR, 2023a.

---

> ### Comment · Reviewer_zjLd · 2024-11-18
>
> I thank the authors for taking the time to respond and revise the manuscript.
>
> I now understand Figure 2 better but still have some doubts about the use of the term "edge detector". If the authors mean something that "only match the specific indicated region", perhaps they shouldn't call them edge detectors. The term edge detector is long-standing in image processing. It typically refers to algorithms like Canny edge detector [1] and Sobel edge detector, which detect edges of different shapes in all regions of an image.
>
> [1] Canny, John. A computational approach to edge detection. TPAMI (1986).

---

> > ### Author Response · Authors · 2024-11-19
> >
> > We thank the reviewer for the quick response.
> >
> > By edge-detector we indeed meant that the eigenvector activates whenever there is an edge at a certain location in the input (not in general). We have changed the language toward “localized edge detector” to specify its behaviour further, thanks for this comment.
> >
> > Is there anything else the reviewer wishes to see in the paper?
> > - If so, please let us know.
> > - If not, we would appreciate it if the reviewer would consider raising their score.

---

> > > ### Comment · Reviewer_zjLd · 2024-11-20
> > >
> > > Thanks for your response. I don't have further questions. I'll be happy to raise score if the correction is implemented (currently Figure 2 caption and Section 4.1 about edge detectors seem unchanged, perhaps the authors forgot to upload their revision?)

---

> > > > ### Author Response · Authors · 2024-11-20
> > > >
> > > > We have updated the manuscript, we had indeed forgotten to upload the revision, our apologies.
> > > > We thank the reviewer for their consideration of our changes.

---

### Official Review · Reviewer_w2jE · 2024-11-01

**Soundness:** 3
**Presentation:** 3
**Contribution:** 4
**Rating:** 8
**Confidence:** 3

**Summary:**

The paper gives several methods to interpret bilinear multilayer perceptron networks (BMLP). BMLP's use a bilinear activation function and produce comparable results to other activation functions. By posing the bilinear leayrs as tensor operations, the author is able to apply direct interaction, eigendecomposition, and SVD analysis. Using these, the author provides an analysis of internal features, construct low-rank approximations of a model, and constructs adversarial examples. Overall, the paper shows the viability of an interpretable BMLPs.

**Strengths:**

Good exhibition of content in all sections, both setup and presentation of results
Effective choice of experiments with convincing examples and helpful visualization.
Work is significant: Presents analysis using well understood methods to produce precise, understandable and actionable interpretations of models with competitive performance.

**Weaknesses:**

I did not notice much analysis using SVD, perhaps I missed it.

**Questions:**

Can you justify the use of the HOSVD as used? Two concerns: 1) The tensor represents a non-linear function. 2) If removed all but but the largest singular values of B, would the resulting tensor functionally approximate B? Also, what were the results of the SVD analysis, what section is that in?


Minor Comment:
Giving some dimensions on W,V could help with the presentation.

---

> ### Author Response · Authors · 2024-11-16
>
> We thank the reviewer for their comments regarding the benefits of our interpretability techniques. We are pleased the reviewer appreciated the significance and presentation of our work, particularly the choice of experiments and examples.
>
> > I did not notice much analysis using SVD, perhaps I missed it.
>
> Given the symmetric nature of all the matrices we consider, we opted to use the eigendecomposition over the SVD (which are similar in nature and equivalent up to sign for symmetric matrices). The eigendecomposition has two appealing qualities over SVD:
> It splits positively and negatively contributing features (according to eigenvalues).
> The projection matrix is the same on each side, making interpretability easier.
>
> > Can you justify the use of the HOSVD as used? Two concerns: 1) The tensor represents a non-linear function. 2) If removed all but but the largest singular values of B, would the resulting tensor functionally approximate B? Also, what were the results of the SVD analysis, what section is that in?
>
> 1) The B tensor is a multilinear map, meaning it is linear in each dimension while holding the others constant. But when we use the same vector x as the input to both of its input dimensions, then the overall output is non-linear in x.
>
> 2) Yes, HOSVD and other approaches for truncating the B tensor will approximate B. They provide a good approximation even when different input vectors x1 and x2 are used instead of the same vector x as in our use case.
>
> 3) We did not sufficiently highlight the HOSVD approach, which we have added to appendix D. This approach combines both SVD (to find the shared output features) and the eigendecomposition (to find the principal interactions for those features).
>
> > Minor Comment: Giving some dimensions on W, V could help with the presentation.
>
> We have also added named dimensions when introducing new matrices in Sec. 2. Thanks for the suggestion.
>
> We look forward to hearing whether the changes are suitable and whether the reviewer has additional follow-ups, suggestions, or questions.

---

> > ### Comment · Reviewer_w2jE · 2024-11-18
> > **Response to Author's Response**
> >
> > We thank the author or their response. No further comments or questions are forthcoming at this time.

---

> > > ### Author Response · Authors · 2024-12-03
> > >
> > > Given that the discussion period has almost ended, we wanted to ask whether the reviewer has had time to review the latest changes to the manuscript. Beyond incorporating the requested changes, we’ve added results to Section 5, verifying our claims on substantially larger models. Furthermore, we generally polished the paper with all the reviewer’s remarks. We appreciate the positive feedback the reviewer has given thus far. We wondered if the reviewer feels the work deserves more awareness, which could be reflected in a score increase.

---

### Official Review · Reviewer_9dUb · 2024-11-03

**Soundness:** 3
**Presentation:** 3
**Contribution:** 2
**Rating:** 8
**Confidence:** 2

**Summary:**

The submission picks up on preliminary ideas developed in “A technical note on bilinear layers for interpretability” (Sharkey, arXiv, 2023) where the idea of using bilinear layers in place of traditional MLPs was promoted to enable forms of interpretability. This is enables by the fact that a bilinear transform of an input vector can be expressed as linear operations with a third-order tensor. This submission follows through by demonstrating several ways in which the tensor can be analyzed with tools from eigen-analysis because input vectors representing features act upon the tensor to yield a matrix (and one can further only consider the symmetric component of this matrix, which has a real eigen-spectrum).

Examples showcased are:
* Interpretability for image classification with a shallow feed-forward network with one hidden bilinear layer (MNIST and FMNIST)
    * Eigenvectors of the last layer when projected back into input space visually resemble class prototypes or key features
    * Top eigenvectors are consistent across training runs and model sizes
    * Success on the mechanistic interpretability challenge (Casper, 2023)
    * Successful adversarial masks created from leading eigenvectors for a chosen adversarial class
* Identifying interactions between sparse autoencoder (SAE) based features in a small language transformer model
    * An example of identifying a sentiment negation circuit by projecting input features onto leading (both positive and negative eigenvalues) eigenvectors
    * Pointing out that SAE features tend to be low-rank in terms of the interaction matrix, suggesting promise for future work in circuit discovery in such models

**Strengths:**

Originality:  While the ideas of using bilinear layers and use of eigenanalysis on the corresponding reduced tensor are borrowed from past work (in particular, “A technical note on bilinear layers for interpretability), the precise demonstrations are new, to my knowledge.

Clarity: The paper is well-written, making it easy to follow (even for a reader like me, less familiar with this area of work). The Discussion section well-summarizes the contributions, contextualizes the implications, and identifies key limitations.

Quality: The work seems technically sound, with a reasonable span of experiments to make the point that’s being made.

Significance: The results are certainly intriguing, and help further the case for adoption of bilinear layers in modern models that are heavily MLP-based.

**Weaknesses:**

The illustrations in the submissions are some fairly reasonable demonstrations of the interpretability-advantages of networks with bilinear layers. However, these ideas may not be mature enough, hence the lack of robust demonstrations at a broader range of tasks, models, and circuits. Being able to showcase such range would significantly strengthen the influence of the work in the nearer future.

Perhaps some of the existing problems can be solved with advances in discovering semantic “output directions” as part of the method. Even if the top eigenvectors may correspond to distributed semantics, further work to enable interpretation of such semantics would also be significant.

**Questions:**

None.

---

> ### Author Response · Authors · 2024-11-16
>
> We thank the reviewer for their insightful comments. We are encouraged to hear that the paper was understandable and helpful for people less familiar with the area.
>
>
> >...hence the lack of robust demonstrations at a broader range of tasks, models, and circuits. Being able to showcase such range would significantly strengthen the influence of the work in the nearer future.
>
> Balancing the clarity of the presentation with the range and depth of demonstrations was challenging. We appreciate any suggestions on the types of demonstrations or tasks that would further strengthen our work.
>
> For image classification, we opted to do a “deep dive” to exhaustively demonstrate the utility and robustness of the features derived from model weights (e.g., truncating the model with minimal accuracy loss, assessing similarity across runs and model sizes, using features to reveal the impact of regularization, etc).
>
> For language modeling (likely the main use-case for our approach), our approach showed that SAE features for the MLP outputs can be computed using surprisingly few MLP input directions and highlighted an interesting example.
>
> Since our submission, we have trained a larger GPT-2 medium-sized model and will add results similar to those in Sec. 4 to the paper to demonstrate that our results are generalized.
>
> > Perhaps some of the existing problems can be solved with advances in discovering semantic “output directions” as part of the method.
>
> Regarding semantic output directions, one approach we did not sufficiently highlight is HOSVD. This allows us to find the most important output directions (in contrast to only picking one). Intuitively, this finds the most important “shared features”. We did not include these experiments as the sharing in these small models is limited and less immediately interpretable than the main results. However, we have added an additional appendix D and added references to it in Sec. 3.3.
>
> On this note, an advantage of bilinear MLPs is that they allow for flexible analysis, particularly using various tensor decompositions. In future work, we plan to develop approaches suitable for analyzing multiple MLP layers simultaneously.
>
> Please let us know if the changes are satisfactory and whether you have further suggestions for improving this work.

---

> ### Author Response · Authors · 2024-11-22
>
> We added additional experiments on GPT2-level models trained on FineWeb (instead of TinyStories) in Section 5, specifically Figure 9. We find our results to generalize to these models that are an order of magnitude larger, strengthening our claims. We also added Appendix H, which discusses that high-quality SAEs are better explained using fewer eigenvectors.
>
> Additionally, in a previous revision, we added Appendix D, which discusses HOSVD to find more ‘semantic’ output directions and may interest the reviewer regarding their raised suggestions.
>
> If the reviewer feels we have not sufficiently addressed their concerns, we would kindly appreciate some targeted feedback. Otherwise, we would be grateful if the reviewer considered increasing their score.

---

> ### Comment · Reviewer_9dUb · 2024-11-28
> **Follow-up to rebuttal**
>
> I thank the authors for their response, and the additional experiments and analysis, and am improving my score to 8 since the follow-up strengthens the paper adequately.

---

### Official Review · Reviewer_QYom · 2024-11-08

**Soundness:** 4
**Presentation:** 3
**Contribution:** 3
**Rating:** 8
**Confidence:** 4

**Summary:**

This work provides an in-depth analysis of Bilinear Multilayer Perceptrons (MLPs).
Bilinear MLPs use bilinear transformations to achieve nonlinearity for function approximation.
The analysis focuses on the spectral decomposition of the bilinear MLP weights.
This reveals the low-rank structure of the weights and provides insight into crafting adversarial samples, overfitting, and so on.
The low-rank structure is interpretable and shows that components corresponding to small eigenvalues can be removed while preserving performance.
The work provides an explainable framework, which would be difficult to achieve without bilinear weights because of the use of nonlinear activation functions.
The paper provides experimental results to support its claims.

**Strengths:**

ORIGINALITY:
This work distinguishes itself from existing approaches to interpretable neural networks by using a unique characteristic of bilinear MLPs. While previous works have utilized bilinear MLP networks, this paper uniquely demonstrates how to exploit their inherent differences from other MLPs to achieve interpretable neural networks.

SIGNIFICANCE:
Interpretable AI is crucial for understanding how AI models encode information.
This work offers a new framework for interpretable neural networks, with the potential to significantly advance the field.

QUALITY:
The quality of this work is good. It provides relevant information to support its claims. The experimental results sufficiently support the main claims in this paper.

CLARITY:
This paper is well-written. It provides enough information required to put this work into proper perspective. It is also well organized and provides enough background information needed to understand the main components of this work.

**Weaknesses:**

This is a minor suggestion:

This paper introduces the use of sparse autoencoders in Section 3.1.
While the stated limitations offer justification for this approach, a more detailed explanation of the role of sparse autoencoders in Section 3.1 would enhance the clarity and comprehensiveness of the work.

**Questions:**

Check weaknesses.

---

> ### Author Response · Authors · 2024-11-16
>
> We are delighted the reviewer shares our enthusiasm for the paper and the proposed approach. We are also happy to hear that the presentation was clear.
>
> > a more detailed explanation of the role of sparse autoencoders in Section 3.1 would enhance the clarity and comprehensiveness of the work
>
> We have expanded our explanation of the use of SAEs in Section 3.1. We clarified that the overall approach can be used for any dictionary of input and output features, but it is not specific to SAEs.  With SAEs, however, the encoder directions can be used to transform the B tensor such that output feature activations can be inferred directly from input activations.
>
> We would strongly appreciate any further feedback (even nitpicks) to strengthen this work further.

---

> > ### Comment · Reviewer_QYom · 2024-11-25
> >
> > Thank you for your response.

---

> ### Author Response · Authors · 2024-12-03
>
> With the discussion period closing, we wondered whether the reviewer was satisfied with the modifications to the paper. In short, we scaled our experiments on language models to GPT2-medium, a 10x scale increase, and showed they generalized; we also added two new appendices (D and H). If the reviewer feels the work is deserving, a score increase would significantly increase the paper’s visibility (and impact).

---

### Author Response · Authors · 2024-12-03

We thank all reviewers for their time and effort. We believe this discussion period has been fruitful, resulting in several changes that significantly strengthened the paper. In short:
- In Section 5, we performed additional experiments on large GPT2-sized models, showing that our findings generalized.
- In Appendix H, we added additional results regarding the impact of SAE training time on our results. We find a form of hidden generalization, where good SAEs have lower-rank feature interactions.
- In Appendix D, we added results regarding using HOSVD as discussed in 3.3. This shows we can find ‘semantic’ output directions from the weights.
- We added additional explanations, such as in subsection 3.1, providing more background and details about our results.
- We have standardized and improved terminology and notation, resulting in more clarity.

We sincerely believe the paper is much more complete and precise and hope the reviewers (as well as others reading this) agree.

---

### Meta-Review · Area_Chair_7gw7 · 2024-12-17

**Metareview:**

The paper explores Bilinear MLPs for their interpretability in deep learning. Bilinear layers are proposed as interpretable alternatives to current functions, offering a viable framework for understanding deep-learning models. The reviewers find the contribution interesting, however I do believe a much more detailed related work should be included in the camera-ready. Currently, there is no discussion about bilinear layers, despite their rich history in both ML and vision, or in fact the tensors and tensor decompositions. I hope the reviewers can rectify the issue - there are multiple papers on the topic and the authors can find even multiple review papers including summarized developments, e.g. the recent [1].

[1] Tensor methods in computer vision and deep learning.

**Additional Comments On Reviewer Discussion:**

The authors raised questions regarding the notation, and I hope this can be further improved in the camera-ready.

---

### Decision · Program_Chairs · 2025-01-22

Accept (Spotlight)